# LLM-based Embeddings:
# Attention Values Encode Sentence Semantics Better Than Hidden States

**Yeqin Zhang** [1 2]  **Yunfei Wang** [1 2]  **Jiaxuan Chen** [1 2]  **Ke Qin** [1 2]  **Yizheng Zhao** [1 2]  **Cam-Tu Nguyen** [1 2]

## Abstract

Sentence representations are foundational to many Natural Language Processing (NLP) applications. While recent methods leverage Large Language Models (LLMs) to derive sentence representations, most rely on final-layer hidden states, which are optimized for next-token prediction and thus often fail to capture global, sentence-level semantics. This paper introduces a novel perspective, demonstrating that attention value vectors capture sentence semantics more effectively than hidden states. We propose *Value Aggregation (VA)*, a simple method that pools token values across multiple layers and token indices. In a training-free setting, VA outperforms other LLM-based embeddings, even matches or surpasses the ensemble-based MetaEOL. Furthermore, we demonstrate that when paired with suitable prompts, the layer attention outputs can be interpreted as aligned weighted value vectors. Specifically, the attention scores of the last token function as the weights, while the output projection matrix ($W_O$) aligns these weighted value vectors with the common space of the LLM residual stream. This refined method, termed *Aligned Weighted VA (AlignedWVA)*, achieves state-of-the-art performance among training-free LLM-based embeddings, outperforming the high-cost MetaEOL by a substantial margin. Finally, we highlight the potential of obtaining strong LLM embedding models through fine-tuning Value Aggregation.

## 1. Introduction

Text representation is fundamental to natural language processing, enabling tasks such as semantic similarity, retrieval, clustering, and anomaly detection (Muennighoff et al., 2023; Enevoldsen et al., 2025; Xiao et al., 2024; Thakur et al., 2021). By embedding text into high-dimensional vector spaces, models can effectively capture semantic relationships and enhance downstream applications such as Retrieval-Augmented Generation (Karpukhin et al., 2020; Gao et al., 2021; Reimers & Gurevych, 2019), among other higher-level tasks (Karpukhin et al., 2020; Gao et al., 2021; Reimers & Gurevych, 2019). Recent studies also exploit internal text representations (embeddings) for model behavior interpretability, showing that attention sinks, compression valleys, and residual-stream representations influence embedding geometry and token-level predictions (Wang et al., 2023; Skean et al., 2025; Queipo-de-Llano et al., 2025; Wendler et al., 2024; Wu et al., 2025).

Despite their strong generative capabilities, autoregressive large language models (LLMs) are not inherently optimized for producing high-quality representations. Trained with a next-token prediction objective, these models differ fundamentally from representation learning approaches that focus on sequence-level semantics. Consequently, autoregressive LLMs often struggle on tasks requiring robust sentence-level embeddings (Muennighoff et al., 2025; BehnamGhader et al., 2024; Zhang et al., 2026; Su et al., 2025).

To address these, existing work follows two main directions. The first introduces additional pretraining objectives to adapt LLMs into embedding models, as in LLM2Vec (BehnamGhader et al., 2024), LLM2Comp (Zhang et al., 2026). However, these methods incur substantial training costs and often compromise the model's generative capabilities. The second direction employs different inference-time strategies such as Explicit One-Word Limitation (EOL) prompting, which concentrates sentence information into the next token and improves representation quality without parameter updates (Lei et al., 2024; Cheng et al., 2025). This approach, however, still exhibits a performance gap compared to fully trained LLM-based embedding models.

A key limitation of many existing approaches lies in their reliance on the last-layer hidden state as the sentence embedding. In this paper, we explore whether alternative internal features of LLMs, beyond hidden states, provide a stronger basis for representing sentence semantics. Addressing this

---

[1]State Key Laboratory for Novel Software Technology, Nanjing University, China [2]School of Artificial Intelligence, Nanjing University, China. Correspondence to: Cam-Tu Nguyen <ncamtu@nju.edu.cn>.

*Proceedings of the 43rd International Conference on Machine Learning*, Seoul, South Korea. PMLR 306, 2026. Copyright 2026 by the author(s).

question has important implications for both training-based and training-free approaches to LLM embeddings.

Inspired by truth conditional semantics (Davidson, 1967), which defines two sentences as semantically similar when their truth values remain consistent across contexts, we propose that sentence meaning can be approximated by treating text continuation space as observable proxies for a sentence's contextual space. We then empirically show that a sentence's influence on its continuation is primarily determined by its value vectors.

Building on this insight, we introduce *Value Aggregation (VA)*, a straightforward approach that aggregates token value vectors across multiple layers and positions. In a training-free setting, VA consistently outperforms existing LLM-based embedding methods and even rivals or exceeds the ensemble-based MetaEOL on MTEB tasks across two base models (LLaMA-2 (7B) and Qwen-3 (8B)). Furthermore, we demonstrate that, with appropriate prompting, the attention layer outputs can be interpreted as aligned weighted VA. Specifically, the last token's attention scores act as weights, while the output projection matrix ($W_O$) maps the weighted values into the shared residual-stream space of the LLM. This enhanced method, termed *Aligned Weighted Value Aggregation (AlignedWVA)*, achieves state-of-the-art performance among training-free LLM-based embeddings, surpassing the high-cost MetaEOL baseline by a substantial margin. It is noteworthy that AlignedWVA incurs significantly lower encoding cost, as it does not require passing each sentence through multiple prompts as in MetaEOL.

Although fine-tuning is not the primary focus of this paper, we show that *Finetune-VA* achieves performance comparable to fine-tuning mean-pooled hidden states while requiring far fewer trainable parameters.

This work makes the following key contributions:

1. **Theoretical insight:** We establish that value vectors, rather than hidden states, more directly capture sentence semantics by encoding a sentence's influence on its continuation under the truth-conditional framework.

2. **Methodology:** We propose *Value Aggregation (VA)*, a simple and training-free method that aggregates value vectors across layers and tokens to produce high-quality sentence embeddings. We further extend VA to *Aligned Weighted Value Aggregation (AlignedWVA)*, which aligns weighted VA with the residual stream space, achieving state-of-the-art performance among training-free LLM embeddings. Note that Aligned-WVA offers substantial computational advantages over ensemble-based methods such as MetaEOL,

3. **Fine-tuning:** We preliminarily demonstrate the potential of developing LLM embedding models through

fine-tuning VA. Additionally, *Finetune-VA*, when applied only to the attention layers, achieves performance comparable to fine-tuned HS baselines while requiring substantially fewer trainable parameters.

## 2. Preliminaries

Given a standard decoder-only Transformer with pre-layer normalization (pre-LN) as in Figure 1, each layer takes the output of the previous layer, performs calculations (Self-Attention and Feed-Forward), and produces a new vector. This vector is the hidden state. Given an input sequence of $N$ tokens, the hidden state $\mathbf{x}_n^l \in \mathbb{R}^d$ represents the representation of token $n$ at layer $l$, where $l \in \{0, 1, ..., L\}$. The initial hidden state $\mathbf{x}_n^0$ is derived from the sum of token and positional embeddings. The model processes the input through $L + 1$ transformer layers (from 0th to $L$th layer).

**Multi-head self-attention (MHA).** At layer $l$, the MHA operation comprises $H$ attention heads, each with head dimension $d_h = d/H$. For head $h \in \{1, ..., H\}$, the query, key, and value vectors for token $n$ are obtained via linear projections of the layer-normalized hidden states:

$$\mathbf{q}_n^{l,h} = \text{LN}(\mathbf{x}_n^{l-1})\mathbf{W}_Q^{l,h},$$
$$\mathbf{k}_n^{l,h} = \text{LN}(\mathbf{x}_n^{l-1})\mathbf{W}_K^{l,h},$$
$$\mathbf{v}_n^{l,h} = \text{LN}(\mathbf{x}_n^{l-1})\mathbf{W}_V^{l,h},$$

where $\mathbf{W}_Q^{l,h}, \mathbf{W}_K^{l,h}, \mathbf{W}_V^{l,h} \in \mathbb{R}^{d \times d_h}$ are learnable projection matrices. The attention weight that token $n$ assigns to token $j$ in head $h$ is computed as:

$$\alpha_{n,j}^{l,h} = \frac{\exp((\mathbf{q}_n^{l,h})^\top \mathbf{k}_j^{l,h}/\sqrt{d_h})}{\sum_{k=1}^{N} \exp((\mathbf{q}_n^{l,h})^\top \mathbf{k}_k^{l,h}/\sqrt{d_h})}.$$

The head attention output for the head $h$ and token $n$ is the weighted sum of value vectors:

$$\mathbf{z}_n^{l,h} = \sum_{j=1}^{N} \alpha_{n,j}^{l,h} \mathbf{v}_j^{l,h} \in \mathbb{R}^{d_h}. \tag{1}$$

The head outputs are concatenated and projected by $\mathbf{W}_O^l \in \mathbb{R}^{d \times d}$ to produce the layer attention output of token $n$:

$$\mathbf{z}_n^l = \text{Concat}\left(\mathbf{z}_n^{l,1}, \ldots, \mathbf{z}_n^{l,H}\right) \tag{2}$$
$$\mathbf{a}_n^l = \text{MHA}^l(\text{LN}(\mathbf{x}_n^{l-1})) = \mathbf{z}_n^l \mathbf{W}_O^l \tag{3}$$

**Feed-forward network (FFN).** The FFN sublayer at layer $l$ processes the attention output through two linear transformations with a non-linear activation:

$$\mathbf{f}_n^l = \text{FFN}^l(\text{LN}(\mathbf{x}_n^{l-1} + \mathbf{a}_n^l))$$
$$= \text{GeLU}((\text{LN}(\mathbf{x}_n^{l-1} + \mathbf{a}_n^l))\mathbf{W}_1^l)\mathbf{W}_2^l.$$

where $\mathbf{W}_1^l \in \mathbb{R}^{d \times d_{\text{ff}}}$ and $\mathbf{W}_2^l \in \mathbb{R}^{d_{\text{ff}} \times d}$ are the weight matrices, with $d_{\text{ff}}$ typically set to $4d$.

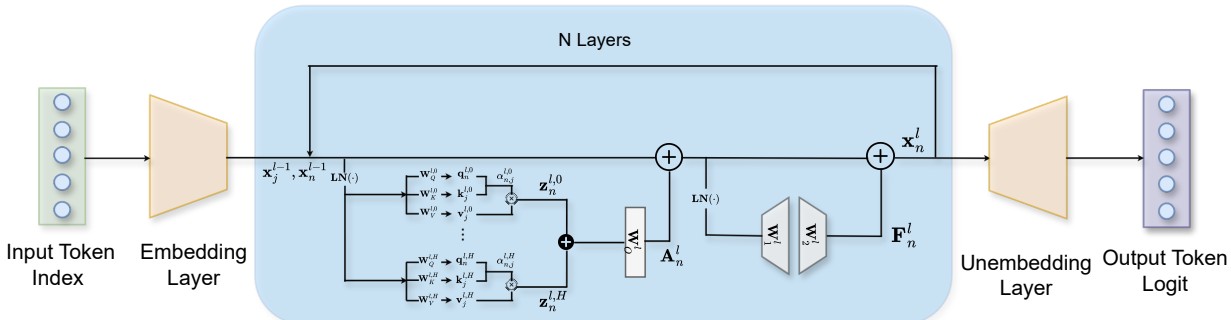

*Figure 1.* The value aggregation architecture, which involves pooling the token-level value representations across all layers.

**Residual stream.** In decoder-only Transformer models, information flows through residual connections. Each token's HS at layer $l$ is computed by adding the outputs of attention and FFN sublayers to the previous layer's HS:

$$\mathbf{x}_n^l = \mathbf{x}_n^{l-1} + \mathbf{a}_n^l + \mathbf{f}_n^l,$$

where $\mathbf{a}_n^l$ and $\mathbf{f}_n^l$ are the outputs of the attention and FFN sublayers respectively.

For LLM-based embeddings, existing methods primarily pool over hidden states $\mathbf{x}_n^l$, with several variants: (1) **LT** uses the hidden state of the last token from the last layer; (2) **WMP** applies weighted mean pooling over the hidden states of the last layer; (3) **HS (Full)** averages hidden states across all layers, while **HS (Half)** averages over the latter half; and (4) **HS** aggregates over a subset of layers chosen based on validation performance. Detailed notations are provided in Table 3. Alternatively, we propose **Value Aggregation (VA)**, which aggregates value vectors $\mathbf{v}_n^{l,h}$ across layers and tokens (see Section 4 for details).

## 3. Motivation

This section shows that HS better aligns with next-token embeddings, while VA is better at capturing sentence semantics within the truth-conditional framework.

### 3.1. Hidden States Align Next Tokens Embedding

In LLMs, the final hidden state at position $t-1$, $\mathbf{x}_{t-1}^L$, is directly optimized to predict the next token $x_t$. This training objective encourages $\mathbf{x}_{t-1}^L$ to encode features that discriminate the correct next token from other vocabulary candidates, rather than features that capture the overall meaning of the entire prefix. Consider two prefixes that share the same local suffix and therefore yield a similar next-token distribution:

$s_1$ : "After months of negotiations, the merger was
approved, and the CEO said the deal was done."

$s_2$ : "After months of recovery, the patient was stable,
and the surgeon said the procedure was done."

Immediately before the final token *done*, both prefixes strongly constrain the set of plausible continuations (e.g., *done*, *complete*, *successful*). Because $\mathbf{x}_{t-1}^L$ is trained to assign high probability to the correct continuation, the representations $\mathbf{x}_{t-1}^L(s_1)$ and $\mathbf{x}_{t-1}^L(s_2)$ may become close in the representation space, even though the overall sentence semantics differ substantially. This can collapse semantically distinct sentences that share similar local continuations, and is therefore not guaranteed to align with sentence similarity. In Appendix D, we show that autoregressive pretraining can be interpreted through the lens of contrastive learning: it pulls the context representation toward the unembedding vector of the observed next token while pushing it away from those of other tokens.

### 3.2. Value Aggregation Encode Sentence Semantics

**Truth-Conditional View of Sentence Similarity.** Truth-conditional semantics defines the meaning of a sentence through the conditions under which it is true in a given context (Davidson, 1967). This framework provides a formal basis for comparing sentence similarity. Building on this concept, we introduce graded truth assessments, where each context is assigned a probability weight representing the likelihood that the sentence is true in that context. This approach provides a more nuanced understanding of sentence meaning, with different contexts supporting varying degrees of truth. Since the set of possible contexts is unobservable, we use sentence continuations as a proxy for truth conditions, assuming that the probability distribution over continuations approximates the likelihood of the corresponding truth conditions. This assumption links continuation probabilities to truth conditions and forms a probabilistic framework for sentence-level semantics. Further details are provided in Appendix B.

**Hypothesis 3.1** (Value Aggregation as a Better Proxy for Sentence Semantics)**.** If the continuation distribution serves as a proxy for truth conditions, then embeddings that more accurately represent this distribution yield better sentence representations. In particular, we hypothesize that value ag-

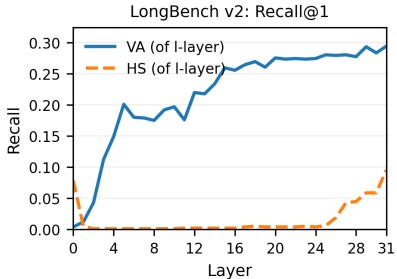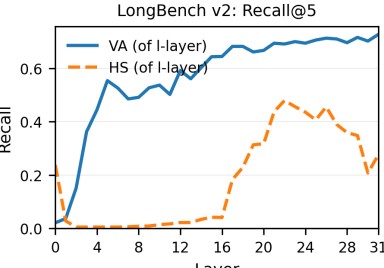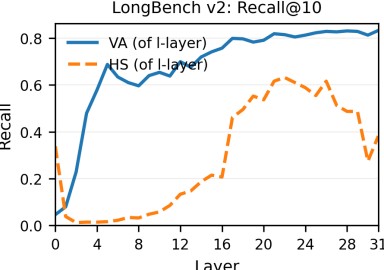

*Figure 2.* **Layer-wise segment matching on LongBench v2.** We split each long sequence into a prefix and suffix segment (split point sampled between $1/4$ and $3/4$ of the token length), and retrieve the matching suffix segment for each prefix segment using embeddings from each layer. Each panel reports recall@k (left: $k=1$, middle: $k=5$, right: $k=10$). VA (of l-layer) improves steadily with depth and outperforms HS (of l-layer) in deeper layers across all $k$, while the hidden state is competitive only in very early layers.

gregation captures continuation distributions more faithfully than hidden-state embeddings. Representations based on value aggregation outperform those based on hidden states in modeling the continuation distribution.

**Empirical Verification.** We test Hypothesis 3.1 using a segment-matching retrieval task. From a long text, we sample a split point uniformly from the middle half of the token positions (between $1/4$ and $3/4$ of the length), producing a *prefix* segment and a *suffix* segment. For each layer, we compute embeddings for all prefix segments and all suffix segments, form a similarity matrix, and evaluate recall@k where the correct match for prefix segment $i$ is suffix segment $i$. We use LongBench v2 because it contains long sequences, ensuring both segments contain enough tokens to carry meaningful content.

Figure 2 presents the layer-wise recall@1/5/10 results for VA and HS. Here, "layer-wise" means that HS (or VA) is obtained by mean-pooling token representations within each layer for comparison. VA performance increases monotonically with depth and consistently surpasses HS in the deeper layers across all three metrics. In contrast, HS achieves higher scores only in the earliest layers but declines rapidly and recovers only partially in later layers. These findings support Hypothesis 3.1, indicating that VA provides a more reliable signal for matching longer continuations than HS.

## 4. Value Aggregation for Text Embeddings

This section first provides a formal description of Value Aggregation (VA), followed by a detailed explanation of the layer selection process for pooling value vectors.

### 4.1. Value Aggregation Method

For each layer $l$ and token $n$, we concatenate the head-level value vectors to obtain a $d$-dimensional token embedding:

$$\mathbf{v}_n^l = \left[ \mathbf{v}_n^{l,1}; \ldots; \mathbf{v}_n^{l,H} \right] \in \mathbb{R}^d.$$

We then summarize the sequence at layer $l$ by mean pooling over tokens:

$$\hat{\mathbf{v}}^l = \frac{1}{N} \sum_{n=1}^{N} \mathbf{v}_n^l \in \mathbb{R}^d.$$

Let $\mathcal{S} \subseteq \{1, \ldots, L\}$ be the set of layers used for aggregation. The VA sentence embedding is:

$$V_{\text{agg}}(x_{1:N}) = \frac{1}{|\mathcal{S}|} \sum_{l \in \mathcal{S}} \hat{\mathbf{v}}^l \in \mathbb{R}^d.$$

It should be noted that VA requires only a standard forward pass to read $\{\mathbf{v}_n^{l,h}\}$ from each attention layer without any extra prompt and calculation. The aggregation itself is mean pooling over tokens and layers, where the layer selection $S$ is selected according to the following section.

### 4.2. Layer Selection for VA

Prior studies on encoder-only LMs reveal a general progression from surface-level features in early layers to higher-level linguistic representations in deeper layers (de Vries et al., 2020; Jawahar et al., 2019). For autoregressive LLMs, de Llano et al. (2025) further observe that strong single-layer sentence embeddings typically emerge near the latter part of the network (around 85% depth). Building on these insights, we conduct an empirical study to identify suitable layers for VA aggregation and use the results to define a default aggregation range $S$.

**Experimental Setup** We evaluate *single-layer* VA by setting $S = \{l\}$ for each layer $l$, and then compute performance using the corresponding embedding for multiple tasks. This yields a performance–depth curve that illustrates how task performance varies across layers. To mitigate selection bias, we ensure that the data used for layer selection does not overlap with the final evaluation test sets (see Section 5). Specifically, we use: (i) the test sets of BiorxivClusteringP2P.v2 and MedrxivClusteringP2P.v2 for clustering; (ii) SciDocsRR validation and AskUbuntuDupQuestions test sets for

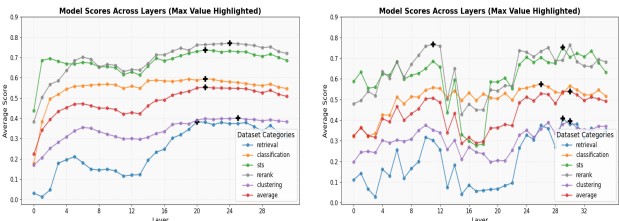

*Figure 3.* Score by Layers for Llama (Left) and Qwen (Right)

reranking; (iii) STS15 test and STSBenchmark validation for semantic textual similarity (STS); (iv) EmotionClassification and Banking77Classification training sets for classification; and (v) SciFact training and NFCorpus validation sets for retrieval. We evaluate our method on two families of LLM backbones: LLaMA-2 and Qwen-3. While LLaMA-2 employs standard multi-head attention, Qwen-3 uses grouped-query attention.

**Results and Analysis**   Figure 3 reports the layer-wise results for LLaMA-2 (7B) and Qwen-3 (8B), with the best-performing layer for each task indicated. Although the optimal single layer varies across tasks, it tends to lie within a relatively narrow range. For LLaMA-2, performance generally increases smoothly with depth, with task-specific peaks emerging in the late-middle layers. In contrast, Qwen-3 shows a less consistent pattern: for most tasks (except classification), performance rises with depth, slightly decreases in the mid-to-late depth, and then increases again near the last layers. Notably, reranking performance peaks much earlier in Qwen-3, whereas deeper layers are more beneficial for retrieval and STS tasks.

These observations highlight two key points. First, relying on a single fixed layer is brittle, as different tasks favor representations from different depths. Second, aggregating a small set of strong layers mitigates sensitivity to any individual layer while retaining the advantages of deep-layer representations. We therefore determine the default layer set $S$ by selecting layers that consistently perform well on the retrieval task (the most challenging task) while excluding those that repeatedly degrade performance across tasks. For LLaMA-2, this selection corresponds to a contiguous deep-middle range (layers *20–27*). For Qwen-3, the strongest layers are concentrated nearer the last layer, and we use layers (*26, 27, 29, 30, and 31*). Unless otherwise specified, VA aggregates value vectors over these model-specific sets.

# 5. Evaluation of VA-based Embeddings

## 5.1. Experimental Setup

**Compared Methods**   We compare our method against several established approaches for sentence representation: LT and WMP (Muennighoff, 2022) use the last token or

weighted mean pooling of the last layer for representation. HS denotes hidden-state pooling baselines defined in Section 2 and summarized in Appendix A. EE (Springer et al., 2025) duplicates the input and applies mean-pooling of the last hidden states in the second copy, allowing the preceding tokens (in the second copy) attend to the later tokens (of the first copy) in causal LMs. PromptEOL (Lei et al., 2024) exploits the prompt *'This sentence: {sentence} means in one word:"'* as a way to fuse sentence semantics to the last token. MetaEOL (Lei et al., 2024) is built on this idea but uses eight ChatGPT-4-designed meta-task prompts. Recently, CP (Cheng et al., 2025) is also built upon PromptEOL by adding an auxiliary prompt to elicit more informative embeddings. We categorize EE, PromptEOL, MetaEOL, and PromptEOL+CP as prompt-based methods, distinguishing them from prompt-free methods such as LT, HS, and WMP. Although they differ in design (with or without prompts), all these baselines use hidden states in different ways.

**Evaluation Datasets**   We evaluate on *14 downstream tasks* across *six dataset groups*, covering clustering, retrieval, semantic textual similarity (STS), classification, and reranking. This setup allows us to assess how well different pooling schemes generalize across a broad set of sentence embedding applications. To ensure that our ablation studies and analyses are not biased toward a particular category or task, this subset was constructed to include tasks from each category in proportions that approximately match those in the full MTEB benchmark. The evaluation metrics followed the MTEB standard (Muennighoff et al., 2023): Accuracy for classification tasks, V-measure for clustering, NDCG@10 for retrieval, MAP for reranking, and Spearman correlation for STS. Detailed information in Table 5 in the Appendix.

**LLM Backbone Models**   We use LLaMA-2 (7B) and Qwen-3 (8B), consistent with Section 4. Notably, due to the grouped-query attention (GQA) mechanism in Qwen-3 8B, value projections are shared across four query groups. As a result, each value vector has one fourth of the hidden-state dimension, corresponding to 1024 dimensions for VA compared to 4096 for HS.

## 5.2. Results and Discussion

**Main Result Analysis**   Table 1 shows the performance of VA in comparison with different baselines. *On average, VA (LLaMA-2) outperforms all prompt-free methods and most prompt-based methods, falling short only to MetaEOL (Qwen-3)*. Notably, VA is a prompt-free method that incurs no additional inference cost, as it does not increase sentence encoding time. In contrast, MetaEOL requires eight forward passes, each corresponding to a distinct prompt, resulting in roughly an $8\times$ higher encoding time compared to VA. The largest gains of VA are observed in retrieval tasks. For

*Table 1.* Performance comparison of different models on a subset of MTEB tasks. "Prompt-free" methods do not append any additional semantic prompt to the input (beyond the standard evaluation instruction), whereas "Prompt-based" methods introduce extra semantic prompts. Our method is prompt-free. MetaEOL is an ensemble that uses eight prompts and therefore requires eight forward passes per example. Echo Embedding (EE) repeats the input sequence and can be viewed as a self-prompting strategy.

| Model | Dim | Backbone | Clustering | | | Retrieval | | | STS | | | Classification | | | Reranking | | Avg. |
|---|---|---|---|---|---|---|---|---|---|---|---|---|---|---|---|---|---|
| | | | Bior. | Medr. | Twen. | SciF. | NFCo. | Argu. | STS17 | SICK-R | STSB. | Bank. | Emot. | Spri. | Stac. | SciD. | |
| | | | | | | | | **Prompt-free Methods** | | | | | | | | | | |
| HS (Full) | 4096 | Llama-2 | 13.69 | 19.86 | 5.67 | 0.10 | 1.43 | 44.11 | 43.90 | 44.46 | 11.18 | 64.24 | **39.34** | 58.41 | 32.38 | 58.69 | 31.25 |
| HS (Half) | 4096 | Llama-2 | 19.86 | 22.68 | 6.30 | 0.23 | 1.46 | 47.8 | 44.21 | 46.64 | 12.15 | 67.58 | 38.67 | 61.83 | 33.04 | 60.79 | 32.37 |
| HS | 4096 | Llama-2 | **20.28** | **23.01** | 6.04 | 0.26 | 1.52 | **47.51** | 44.17 | 46.46 | 12.33 | 66.46 | 38.93 | **62.89** | 33.03 | **61.12** | 33.14 |
| HS (Full) | 4096 | Qwen-3 | 7.42 | 17.47 | 5.27 | 0.17 | 1.58 | 40.71 | 61.05 | 38.97 | 23.54 | 58.95 | 34.85 | 46.83 | 28.41 | 53.08 | 29.88 |
| HS (Half) | 4096 | Qwen-3 | 11.55 | 19.57 | 5.77 | 0.67 | 1.75 | 43.27 | 62.53 | 40.94 | 27.84 | 60.61 | 34.69 | 53.14 | 29.25 | 54.98 | 31.90 |
| HS | 4096 | Qwen-3 | 12.87 | 20.73 | 5.72 | 0.71 | 1.87 | 42.50 | **64.11** | 42.54 | 28.06 | 61.61 | 34.62 | 56.40 | 31.16 | 58.10 | 32.93 |
| LT | 4096 | Llama-2 | 15.99 | 17.42 | **15.96** | 2.17 | 1.31 | 14.24 | 57.8 | 55.63 | 45.72 | 68.65 | 29.85 | 47.01 | 32.07 | 58.83 | 33.05 |
| WMP | 4096 | Llama-2 | 19.73 | 19.47 | 14.54 | **38.89** | **6.13** | 33.59 | 63.91 | **57.52** | **58.01** | 66.42 | 30.97 | 58.48 | **37.74** | 61.05 | **40.46** |
| LT | 4096 | Qwen-3 | 13.56 | 16.22 | 13.57 | 5.06 | 1.57 | 8.42 | 38.56 | 41.48 | 28.53 | 55.82 | 29.80 | 9.77 | 25.25 | 47.35 | 23.93 |
| WMP | 4096 | Qwen-3 | 9.44 | 15.82 | 8.19 | 5.15 | 1.41 | 22.12 | 51.10 | 44.17 | 34.08 | 60.52 | 30.01 | 28.67 | 32.04 | 49.74 | 28.03 |
| | | | | | | | | **Prompt-based Methods** | | | | | | | | | | |
| EE | 4096 | Llama-2 | 22.94 | 23.15 | 25.74 | 25.61 | 9.97 | 25.24 | 80.51 | 70.18 | 71.94 | 81.79 | 45.00 | 68.48 | 40.79 | 60.15 | 46.54 |
| PrompEOL | 4096 | Llama-2 | 22.49 | 21.14 | 31.47 | 27.16 | 13.59 | 11.65 | 79.67 | 73.82 | 75.32 | 76.37 | 47.13 | 26.08 | 37.65 | 66.22 | 43.55 |
| MetaEOL | 4096 | Llama-2 | **30.95** | 26.56 | **40.03** | 40.59 | 16.41 | 21.75 | **82.29** | 76.88 | 76.87 | 82.26 | 51.05 | 48.24 | 39.87 | **77.91** | 50.83 |
| PrompEOL + CP | 4096 | Llama-2 | 22.72 | 21.26 | 28.98 | 33.42 | 17.43 | 14.57 | 80.93 | 72.69 | 74.73 | 77.51 | 47.70 | 28.42 | 37.54 | 66.55 | 44.60 |
| PromptEOL | 4096 | Qwen-3 | 23.43 | 23.57 | 27.70 | 18.27 | 4.56 | 12.12 | 72.84 | 67.98 | 67.80 | 70.06 | 47.22 | 36.49 | 38.91 | 73.85 | 41.77 |
| MetaEOL | 4096 | Qwen-3 | 29.21 | **27.01** | 36.74 | **47.59** | 12.90 | **30.81** | 80.72 | 74.24 | 71.65 | 81.90 | **52.55** | **73.88** | 41.41 | 77.53 | **52.72** |
| PrompEOL + CP | 4096 | Qwen-3 | 27.66 | 24.67 | 34.76 | 27.35 | **18.13** | 14.84 | 74.14 | 69.38 | 73.22 | 71.14 | 48.66 | 22.15 | 40.60 | 75.90 | 44.47 |
| | | | | | | | | **Our Methods** | | | | | | | | | | |
| VA (Full) | 4096 | Llama-2 | 31.69 | 28.30 | 26.34 | 51.28 | 21.00 | 42.75 | 74.51 | 61.47 | 60.94 | 73.89 | 39.58 | 71.42 | 41.63 | 76.16 | 50.07 |
| VA (Half) | 4096 | Llama-2 | 32.45 | 28.65 | 27.95 | 52.41 | 23.52 | 44.26 | 74.08 | 61.49 | 61.72 | 75.19 | 39.54 | 73.75 | 41.51 | 76.70 | 50.94 |
| VA | 4096 | Llama-2 | **33.13** | **29.56** | 30.59 | 54.58 | **25.89** | **45.76** | 75.37 | 61.92 | 63.03 | 76.15 | **39.85** | 75.97 | 42.08 | **77.59** | 52.25 |
| VA (Full) | 1024 | Qwen-3 | 26.85 | 24.65 | 20.59 | 58.37 | 18.69 | 41.41 | 71.70 | 60.38 | 60.77 | 75.72 | 33.20 | 81.92 | 44.72 | 70.45 | 49.24 |
| VA (Half) | 1024 | Qwen-3 | 26.94 | 24.29 | 21.92 | 58.29 | 18.80 | 41.49 | 71.83 | 60.47 | 60.91 | 75.71 | 32.91 | **82.10** | 44.79 | 70.44 | 49.35 |
| VA | 1024 | Qwen-3 | 31.46 | 26.61 | 25.35 | **58.93** | 21.84 | 43.11 | **76.28** | **62.98** | **63.24** | **76.49** | 35.65 | 81.29 | **45.51** | 73.48 | 51.59 |

instance, while MetaEOL achieves the best overall performance on SciFact, VA surpasses it by 10 points on the same task. In semantic textual similarity (STS), prompt-based methods still hold an advantage; however, VA, as a prompt-free method, substantially narrows the gap. For example, among prompt-free baselines, WMP achieves the highest STS performance, yet VA exceeds it across all STS datasets, with more than a 10-point improvement on STS17. These conclusions also hold for Qwen-3, indicating that the effect generalizes across different backbone families and attention mechanisms. Notably, VA (Qwen-3) achieves highly competitive performance while having only one-fourth of the dimensionality, making it substantially more efficient in both sentence encoding and similarity computation.

**VA Layer Selection Analysis** Across both Qwen-3 and LLaMA-2, selecting a subset of layers using the strategy described in Section 4 consistently enhances VA performance. Aggregating from deeper layers (VA (**Half**)) also tends to outperform pooling over all layers (VA (**Full**)). With the selected-layer configuration, VA achieves an average improvement of about 2 points compared to full-layer aggregation (VA (**Full**)) on both backbones, with the largest gains observed in clustering and retrieval tasks. Restricting aggregation to the last half of layers produces a smaller but generally positive effect with both LLM backbones.

## 6. Alternative Strategies for Value Aggregation

### 6.1. From VA to Weighted VA (WVA)

Our default VA applies mean pooling over tokens. This design is simple and parameter-free, but prior work shows that token weighting can substantially influence embedding quality (Muennighoff, 2022). This motivates the *Weighted VA* (WVA) variant, in which aggregation assigns non-uniform importance to tokens. This section reports downstream performance, while Appendix E provides a probing-based interpretation of why weighting helps: when token weights align with attention patterns that support continuation, weighted pooling of value vectors better preserves information predictive of subsequent tokens.

**Weighting Strategies** A simple weighting strategy is to use the attention weights $\alpha_{n,j}^{l,h}$ produced by the last token $\mathbf{x}_n$, which describe how the last token attends to each prefix token. Under standard multi-head attention, this weighted value aggregation is exactly the head-wise output for the last token, and its concatenated form corresponds to $\mathbf{z}_n^l$ in Equations 1 and 2. For models with grouped-query attention (GQA), the number of query heads and key-value heads may differ. We therefore compute token-specific weights by mean-pooling the last-token attention weights across query heads, and then multiply the prefix value vectors by these mean-pooled weights. We refer to this method as *Weighted*

*Table 2.* Performance of alternative value aggregation strategies evaluated on various tasks from the MTEB benchmark.

| Model | Dim | Backbone | Clustering | | | Retrieval | | | STS | | | Classification | | | Reranking | | Avg. |
|---|---|---|---|---|---|---|---|---|---|---|---|---|---|---|---|---|---|
| | | | Bior. | Medr. | Twen. | SciF. | NFCo. | Argu. | STS17 | SICK-R | STSB. | Bank. | Emot. | Spri. | Stac. | SciD. | |
| **Baselines** | | | | | | | | | | | | | | | | | |
| **PrompEOL** | 4096 | Llama-2 | 22.49 | 21.14 | 31.47 | 27.16 | 13.59 | 11.65 | 79.67 | 73.82 | **75.32** | 76.37 | 47.13 | 26.08 | 37.65 | 66.22 | 43.55 |
| **VA** | 4096 | Llama-2 | **33.13** | **29.56** | 30.59 | 54.58 | **25.89** | 45.76 | 75.37 | 61.92 | 63.03 | 76.15 | 39.85 | 75.97 | 42.08 | **77.59** | 52.25 |
| **VA (PromptEOL)** | 4096 | Llama-2 | 32.04 | 28.20 | 33.00 | 49.90 | 8.75 | **46.61** | 79.03 | 65.63 | 63.58 | 77.00 | 44.46 | 77.42 | 41.29 | 76.40 | 51.67 |
| **MetaEOL** | 4096 | Qwen-3 | 29.21 | 27.01 | **36.74** | 47.59 | 12.90 | 30.81 | **80.72** | **74.24** | 71.65 | **81.90** | **52.55** | 73.88 | 41.41 | 77.53 | **52.72** |
| **VA** | 1024 | Qwen-3 | 31.46 | 26.61 | 25.35 | **58.93** | 21.84 | 43.11 | 76.28 | 62.98 | 63.24 | 76.49 | 35.65 | **81.29** | **45.51** | 73.48 | 51.59 |
| **Weighted Value Aggregation** | | | | | | | | | | | | | | | | | |
| **WVA (LT)** | 4096 | Llama-2 | 15.83 | 16.33 | 11.48 | 19.75 | 3.04 | 16.39 | 63.58 | 57.01 | 58.69 | 62.28 | 29.31 | 13.68 | 28.78 | 55.62 | 32.27 |
| **WVA (PromptEOL)** | 4096 | Llama-2 | 27.40 | 22.14 | 30.67 | 45.32 | **24.22** | 31.27 | 85.02 | 73.98 | 77.11 | 81.15 | 53.66 | 57.21 | **43.07** | 75.44 | 51.98 |
| **WVA (FutureEOL)** | 4096 | Llama-2 | 30.46 | 25.21 | 35.04 | 51.80 | 21.54 | 37.95 | 85.04 | 72.60 | 74.64 | 81.54 | 53.86 | 75.83 | 42.86 | **77.22** | 54.69 |
| **WVA (LT)** | 1024 | Qwen-3 | 25.56 | 21.72 | 18.51 | 6.44 | 8.21 | 13.82 | 52.41 | 64.98 | 42.02 | 58.88 | 35.89 | 15.54 | 33.42 | 63.43 | 32.92 |
| **WVA (PromptEOL)** | 1024 | Qwen-3 | 26.18 | 23.71 | 29.30 | 14.70 | 15.80 | 29.63 | 79.32 | 70.41 | 72.33 | 79.67 | 49.96 | 33.95 | 41.36 | 63.99 | 45.02 |
| **WVA (FutureEOL)** | 1024 | Qwen-3 | 26.22 | 23.96 | 37.14 | 24.82 | 17.17 | 26.85 | 80.14 | 71.07 | 73.83 | 75.79 | 48.11 | 22.15 | 42.06 | 69.98 | 45.66 |
| **Aligned Weighted Value Aggregation** | | | | | | | | | | | | | | | | | |
| **AlignedWVA (PromptEOL)** | 4096 | Llama-2 | 28.82 | 23.45 | 31.20 | 45.31 | 25.13 | 32.44 | 83.33 | 73.80 | 76.74 | 82.09 | 52.68 | 60.93 | 43.52 | 75.76 | 52.51 |
| **AlignedWVA (FutureEOL)** | 4096 | Llama-2 | 31.25 | 26.39 | 33.82 | 51.40 | 25.32 | 39.67 | 83.38 | 71.54 | 73.36 | **82.62** | 52.78 | **78.42** | 43.39 | 76.44 | 54.98 |
| **AlignedWVA (PromptEOL)** | 4096 | Qwen-3 | **35.56** | **29.73** | **48.04** | 55.44 | 31.55 | 34.53 | 83.06 | **76.14** | **77.48** | 81.52 | **54.18** | 59.54 | **45.88** | **83.01** | 56.83 |
| **AlignedWVA (FutureEOL)** | 4096 | Qwen-3 | 33.62 | 28.03 | 44.51 | **62.98** | **32.04** | **42.88** | **84.31** | 68.55 | 75.53 | 82.06 | 51.75 | 69.50 | 43.26 | 76.79 | **56.84** |

*Value Aggregation using Last Token*, denoted as **WVA (LT)**.

Directly using the head attention output $\mathbf{z}_n^l$ of the last token as a sentence representation suffers from the same limitation as using the last token's hidden state (LT): the final token alone does not capture the full-sentence semantics. Inspired by PromptEOL, we wrap the sentence with the PromptEOL prompt template, thereby encouraging the model to fuse sentence-level information into the final token. We then use $\mathbf{z}_n^l$ as the sentence representation. This approach is referred to as **WVA (PromptEOL)**.

Inspired by Section 3.2, which shows that sentence semantics are determined by continuation, we propose *FutureEOL*, which aims to capture the continuation of a sentence using the prompt *'Forecasting the subsequent tokens: {sentence} in one word:"'*. We then apply this prompt within the WVA framework, referring to this method as **WVA (FutureEOL)**.

**Experimental Results and Discussion**   Table 2 presents the experimental results. As shown, WVA (FutureEOL) with LLaMA-2 surpasses the strong MetaEOL baseline by 2 points on average. Compared with WVA (LT), which assigns weights to the value vectors of all tokens based on next-token prediction, WVA (PromptEOL)—which explicitly focuses the model on predicting the next token through a prompt—achieves an overall improvement of nearly 20 points, highlighting the importance of using meaningful prompts in WVA. In nearly all tasks, WVA (PromptEOL) outperforms WVA (LT) by about 10 points. Furthermore, transitioning from WVA (PromptEOL) to WVA (FutureEOL) yields additional gains, particularly on clustering and retrieval datasets (except *NFCor*). On STS datasets, WVA (FutureEOL) shows a slight decrease, while in classification tasks, it brings modest improvements on *Bank* and *Emot* but a substantial gain on *Spri*.

## 6.2. From WVA to Aligned WVA

WVA with prompting operates in the attention value space (concatenated across heads), whereas many baselines—including EOL-style representations—operate in the model's residual-stream space. To align these spaces, we introduce an *Aligned WVA* variant that takes the signal after the output projection $\mathbf{W}_O^l$, which maps the weighted aggregation back to the residual stream, as defined in Equation 3. We further derive different variants of AlignedWVA using *PromptEOL* and *FutureEOL*.

**Results and Discussion**   AlignedWVA improves upon WVA for both LLaMA-2 and Qwen-3 under both PromptEOL and FutureEOL. As expected, the alignment provides a substantially larger benefit on Qwen-3, improving performance by over 10 points on average for both prompts. Notably, FutureEOL with AlignedWVA surpasses the ensemble-based MetaEOL baseline on both backbones by a considerable margin. In particular, the best-performing variant, AlignedWVA (FutureEOL) with Qwen-3, exceeds MetaEOL with Qwen-3 by more than 4 points on average. It is noteworthy that AlignedWVA incurs significantly low cost for encoding, as it does not need to pair each encoded sentence with different prompts as in MetaEOL.

## 7. Training LLM Embedding Models by Finetuning VA

Recent LLM-based embedding models (e.g., LLM2Vec (BehnamGhader et al., 2024)) are typically trained by fine-tuning hidden-state representations using contrastive learning. To examine whether contrastive fine-tuning can similarly enhance VA embeddings, we fine-tune the same LLama-2 (7B) backbone with LoRA, applying an identical set of target modules for both VA and the hidden-state

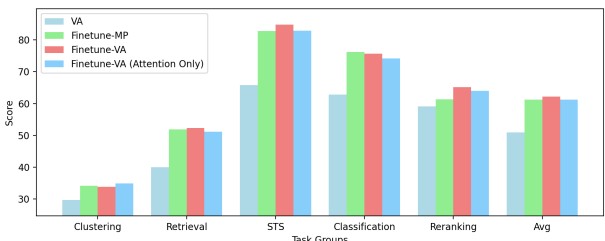

*Figure 4.* Results on the "Average of Major Task Categories" with VA and other finetuned models (LLama-2 (7B) backbone)

mean-pooling baseline (**Finetune-MP**). Specifically, we apply LoRA to the self-attention projections $\mathbf{W}_Q^l, \mathbf{W}_V^l, \mathbf{W}_O^l$ and the feed-forward projections $\mathbf{W}_1^l, \mathbf{W}_2^l$. Under this controlled setup, the only distinction between Finetune-VA and Finetune-MP lies in the pooling operator used to obtain the sentence embedding: Finetune-VA pools attention value vectors across tokens and layers, whereas Finetune-MP performs mean pooling over last-layer hidden states.

We also evaluate a lighter variant, **Finetune-VA (Attention Only)**, which applies LoRA exclusively to the self-attention modules $\mathbf{W}_Q^l, \mathbf{W}_K^l, \mathbf{W}_V^l, \mathbf{W}_O^l$ and the corresponding layer-normalization weights. This configuration reduces the number of trainable parameters to less than one quarter of that in **Finetune-MP**, while keeping all other training settings identical.

**Training Objective**   We perform supervised contrastive learning using annotated positive pairs. Negative samples are constructed via in-batch sampling and hard-negative sampling, where hard negatives are pre-selected using the E5 data with a cross-encoder model (Wang et al., 2024). Following LLM2Vec, we use 1,024,000 training examples drawn from the public portions of its training datasets.

**Results and Discussion**   Figure 4 shows that Finetune-VA improves VA embeddings and yields consistent gains over Finetune-MP. The improvements are greatest on reranking and STS, with gains of 3.8 and 2.02 points, respectively. We observe small degradations on clustering and classification task, suggesting that value-based training primarily benefits tasks that rely on fine-grained semantic matching, while the effect is weaker for category-level discrimination. Finetune-VA (Attention Only) can achieve almost a consistent effect only training 1/4 of the parameters, which suggests the advantages of finetuning VA. *Future research should focus on developing more effective training strategies to fully exploit the representation advantages of VA.*

## 8. Related Works

Large language models generate contextualized representations at multiple layers and token positions. To obtain

fixed-length embeddings, aggregation is typically performed along both the layer dimension and the sequence dimension.

**Layer-wise aggregation of hidden states**   Different layers capture information at varying levels of granularity, with earlier layers focusing on local lexical and syntactic features, while deeper layers optimize pre-training objectives. Consequently, intermediate layers are often more suitable for embedding tasks like retrieval and clustering (Queipo-de-Llano et al., 2025; Stankevičius & Lukoševičius, 2024; Vu et al., 2022). Methods like MLTP (Tang & Yang, 2024), MoEE (Li & Zhou, 2025), and LMORT (Sun et al., 2024) explore multi-layer aggregation, using pooling and Mixture-of-Experts routing weights for improved embeddings.

**Sequence-level aggregation**   Once layers are selected, the next step is to aggregate sequence- or token-level representations. Common methods include mean pooling (Ni et al., 2022), weighted mean pooling (Muennighoff, 2022), and pooling over special tokens like [CLS] in encoder-only models (Reimers & Gurevych, 2019) and [EOS] in decoder-only models (Muennighoff, 2022). Training-based approaches like NV-Embed-v1 (Lee et al., 2025) introduce learnable pooling layers that aggregate token-level information.

**Prompt-based aggregation**   Prompt-based methods focus information by using instruction-style templates. PromptEOL (Lei et al., 2024) and Echo embeddings (Springer et al., 2025) aggregate information from specific tokens. MetaEOL (Lei et al., 2024) and CP (Cheng et al., 2025) further enhance sentence representations using meta-task prompts and auxiliary instructions.

While the methods above primarily rely on hidden states for sentence embeddings, our work takes a novel approach by exploring other internal signals from LLMs, aiming to enrich embeddings beyond traditional hidden-state ones.

## 9. Conclusion

In this work, we revisited the foundation of sentence representation in large language models (LLMs) and demonstrated that value vectors—rather than hidden states—serve as a stronger basis for capturing sentence-level semantics. Building on this insight, we proposed Value Aggregation (VA), a simple yet effective method that aggregates value vectors across layers and tokens to form powerful training-free embeddings. We further introduced Aligned Weighted Value Aggregation (AlignedWVA), which aligns weighted value vectors with the residual stream, achieving state-of-the-art performance among training-free LLM-based embeddings while maintaining low computational cost. Finally, we showed a method for fine-tuning VA that can yield competitive results to fine-tuned mean-pooled HS baselines with

far fewer parameters.

Our findings reveal that the value space of LLMs encodes rich semantic information that has been largely overlooked. We believe this work will inspire further research into value-based representations and their potential to bridge generative modeling and representation learning.

## Acknowledgements

This work was supported by the National Natural Science Foundation of China (Grant No. W2532049).

## Impact Statement

This paper presents work whose goal is to advance the field of Machine Learning, with a particular focus on improving sentence representations derived from large language models. The proposed value aggregation method may benefit applications such as semantic search, retrieval, clustering, and text understanding by improving representation quality and reducing embedding dimensionality in some settings. Like other representation learning methods, it may also inherit biases, privacy risks, or misuse risks from the underlying language models and downstream deployment contexts. Our work does not introduce new user data collection, decision-making systems, or application-specific deployment pipelines. We believe that the broader societal impacts are largely aligned with those of general-purpose representation learning and large language model research.

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

# A. Notation Information

The primary symbols for VA and HS are listed in Table 3, and the main symbols for Transformer layer components are provided in Table 4. Additional notations are as follows:

**Q, K, V Parameters Concatenation.** The following expressions represent the concatenation of the parameters across heads:

$$\mathbf{W}_Q^l = \text{Concat}\left(\mathbf{W}_Q^{l,1}, \dots, \mathbf{W}_Q^{l,H}\right),$$
$$\mathbf{W}_K^l = \text{Concat}\left(\mathbf{W}_K^{l,1}, \dots, \mathbf{W}_K^{l,H}\right),$$
$$\mathbf{W}_V^l = \text{Concat}\left(\mathbf{W}_V^{l,1}, \dots, \mathbf{W}_V^{l,H}\right).$$

**Last Layer Hidden-state embedding (LHS) baselines.** A common approach to obtain a sequence-level representation is to pool the last-layer hidden states across tokens. For instance, mean pooling and last-token pooling are defined as:

$$\mathbf{LHS}_{\text{mean}}(x_{1:N}) = \frac{1}{N} \sum_{n=1}^{N} \mathbf{x}_n^L,$$
$$\mathbf{LHS}_{\text{last}}(x_{1:N}) = \mathbf{x}_N^L.$$

These methods aggregate information that has been processed through the entire network, combining contributions from both attention and FFN operations across all layers.

**Full-layer selection.** The **Full** strategy aggregates representations from all transformer layers. The full-layer representation is defined as:

$$\mathbf{Full}(x_{1:N}) = \frac{1}{L} \sum_{l=1}^{L} \left(\frac{1}{N} \sum_{n=1}^{N} \mathbf{x}_n^l\right).$$

This strategy preserves information from all depths of the network, including both lower-layer surface features and higher-layer semantic features.

**Half-layer selection.** The **Half** strategy aggregates representations only from the deep half of the transformer layers. Let $\mathcal{L}_{\text{half}} = \{L/2, \dots, L\}$ denote the selected layers. The representation is defined as:

$$\mathbf{Half}(x_{1:N}) = \frac{2}{|\mathcal{L}|} \sum_{l \in \mathcal{L}_{\text{half}}} \left(\frac{1}{N} \sum_{n=1}^{N} \mathbf{x}_n^l\right).$$

By discarding lower-layer representations, this strategy focuses on deeper-layer features while reducing the influence of early-layer signals.

**Layer selection baselines.** Layer selection determines which transformer layers contribute to a sequence-level representation. A common design choice is whether to aggregate representations from all layers or only from deeper layers, based on the observation that deeper layers tend to encode higher-level semantic information. We consider two standard layer selection strategies.

# B. Details for Truth-Condition Semantic

**Definition B.1** (Sentence-Level Semantics). Let $X$ denote a set of possible contexts. The meaning of a sentence $s$ is represented by a truth function.

$$[\![s]\!] : X \to \{0, 1\},$$

where $[\![s]\!](x) = 1$ indicates that $s$ is true in context $x$. The truth set of $s$ is

$$T(s) = \{x \in X \mid [\![s]\!](x) = 1\}.$$

To make "truth conditions" concrete, consider a context $x$ that specifies facts such as (i) the location of a particular key, (ii) whether it is the correct key for a given door, and (iii) whether access constraints are satisfied. For the sentence

$$s = \text{"The lab key is in my pocket."},$$

The truth set $T(s)$ contains exactly those contexts where the lab key's location is *pocket*. In contrast, the sentence

$$s' = \text{"I can open the lab door."}$$

has a different truth set, since it additionally depends on whether the key is correct, whether the door is locked, and whether entry is permitted. This highlights that truth conditions are context constraints, rather than informal consequences of an utterance.

**Graded Truth Assessments.** To capture graded truth assessments, we extend this framework by assigning a probability distribution over contexts. Each $x \in X$ is associated with a weight $w_x \in [0, 1]$, which represents the likelihood that the sentence $s$ is true in context $x$. These weights are normalized to sum to 1 across all contexts, ensuring that the total probability mass is properly distributed:

$$\sum_{x \in X} w_x = 1.$$

We define the probabilistic truth function $[\![s]\!]_p : X \to [0, 1]$, which assigns a probability to each context, representing the likelihood that the sentence $s$ is true in that context. This provides a more nuanced understanding of sentence meaning, where different contexts can support varying degrees of truth.

*Table 3.* Notations for description for VA and HS.

| Symbol | Description |
|---|---|
| **HS (Full)** | Pooling hidden state across all layers and all tokens. |
| **HS (Half)** | Pooling hidden state across deep half layers and all tokens. |
| **HS** | Pooling hidden state across selected layers (same as VA) and all tokens. |
| **VA (Full)** | Pooling value vectors across all layers and all tokens. |
| **VA (Half)** | Pooling value vectors across deep half layers and all tokens. |
| **VA** | Pooling value vectors across selected layers (seen in Section 4) and all tokens. |
| **WVA (LT)** | Directly pooling the last token's attention head output $\mathbf{z}_n^{l,h}$ across selected layers to get the weight VA result. |
| **WVA (PromptEOL)** | Pooling the last token's attention head output $\mathbf{z}_n^{l,h}$ of PromptEOL across selected layers to get the weight VA result. |
| **AlignedWVA (PromptEOL)** | Pooling the last token's attention layer output $\mathbf{a}_n^l$ of PromptEOL across selected layers to get the align VA result. |
| **AlignedWVA (FutureEOL)** | Pooling the last token's attention layer output $\mathbf{a}_n^l$ of FutureEOL across selected layers to get the align VA result. |

*Table 4.* Notations for Transformer layer components.

| Symbol | Description |
|---|---|
| $\mathbf{x}_n^l \in \mathbb{R}^d$ | Hidden state of token $n$ at layer $l$. |
| $\mathbf{q}_n^{l,h}, \mathbf{k}_j^{l,h}, \mathbf{v}_j^{l,h} \in \mathbb{R}^{d_h}$ | Query/key/value vectors for token $n/j$ in head $h$ at layer $l$. |
| $\alpha_{n,j}^{l,h}$ | Attention weight from token $n$ to $j$ in head $h$ at layer $l$. |
| $\mathbf{z}_n^{l,h} \in \mathbb{R}^{d_h}$ | Output of attention head $h$ for token $n$ at layer $l$. |
| $\mathbf{W}_Q^{l,h}, \mathbf{W}_K^{l,h}, \mathbf{W}_V^{l,h} \in \mathbb{R}^{d \times d_h}$ | Projection matrices for queries/keys/values in head $h$ at layer $l$. |
| $\mathbf{W}_O^l \in \mathbb{R}^{d \times d}$ | Output projection matrix for multi-head attention at layer $l$. |
| $\mathbf{W}_1^l, \mathbf{W}_2^l \in \mathbb{R}^{d \times d_{\text{ff}}}$ | Weight matrices of the feed-forward network (FFN) at layer $l$ ($d_{\text{ff}} = 4d$ typically). |
| $\mathbf{a}_n^l \in \mathbb{R}^d$ | Multi-head attention output for token $n$ at layer $l$. |
| $\mathbf{f}_n^l \in \mathbb{R}^d$ | FFN output for token $n$ at layer $l$. |
| $\text{LN}(\cdot)$ | Layer normalization operation (applied before each sublayer). |
| $\text{MHA}^l(\cdot)$ | Multi-head self-attention operation at layer $l$. |
| $\text{FFN}^l(\cdot)$ | Feed-forward network operation at layer $l$. |

**Definition B.2** (Probabilistic Truth Function). Given a weight function $w_x$ for each context $x \in X$, we define the probabilistic truth function for sentence $s$ as:

$$[\![s]\!]_p(x) = w_x, \quad \forall x \in X,$$

where $w_x$ represents the likelihood that the sentence $s$ is true in context $x$.

In practice, the set of possible contexts $\Omega$ is unobservable, and directly computing $T(s)$ is often infeasible. However, in natural language processing, we can leverage the idea of sentence continuations as a proxy for truth conditions. Specifically, each continuation corresponds to a distinct truth condition, and the probability of a continuation reflects the likelihood of the associated truth condition.

**Assumption B.3** (Likelihood Approximation of Truth Conditions). We assume that the probability distribution over sentence continuations can approximate the likelihood of

the corresponding truth conditions. Specifically, each continuation corresponds to a distinct truth condition, and the probability of a continuation is proportional to the weight $w_x$ associated with the truth condition it represents. This assumption links the probabilistic interpretation of sentence continuation with the underlying truth conditions that govern sentence meaning.

## C. Dataset Information

The dataset of MTEB subset information can be seen in Table 5.

## D. Hidden States Align Next-Token Embeddings

From a contrastive learning view, autoregressive pretraining induces a token-level discrimination objective, pulling the

_Table 5._ Statistics of evaluation datasets

| Category | Dataset | #Samples |
|---|---|---|
| Clustering (3) | BiorxivCS2S | 75000 |
| | MedrxivS2S | 37500 |
| | TwentyNewsgroups | 59545 |
| Retrieval (3) | SciFact | 5483 |
| | NFCorpus | 3956 |
| | ArguAna | 10080 |
| STS (3) | STS17 | 5692 |
| | SICK-R | 19854 |
| | STSBenchmark | 2758 |
| (Pair) Classification (3) | Banking77 | 3696 |
| | EmotionClassification | 2096 |
| | SprintDuplicateQuestions | 8931 |
| Reranking (2) | StackOverflow DupQuestions | 82798 |
| | SciDocsRR | 89131 |
| Overall | 14 datasets | 406520 |

context representation toward the embedding of the true next token and away from embeddings of other vocabulary tokens. This mechanism explains why pooling hidden states into a single vector inherits a representation space shaped by next-token prediction.

**Definition D.1** (Autoregressive next-token prediction and token-level NLL). Consider the pre-LN Transformer defined above with a causal attention mask. For a token prefix $x_{<t} = (x_1, \ldots, x_{t-1})$, let $\mathbf{v}_x \in \mathbb{R}^d$ denote the unembedding vector of token $x \in V$.

$$p(x_t \mid x_{<t}) = \frac{\exp\big((\mathbf{x}_{t-1}^L)^\top \mathbf{v}_{x_n}/\tau\big)}{\sum_{x \in V} \exp\big((\mathbf{x}_{t-1}^L)^\top \mathbf{v}_x/\tau\big)}.$$

The token-level negative log-likelihood (NLL) is

$$\mathcal{L}_{\text{NLL}}(x_t, x_{<t}) = -\log p(x_t \mid x_{<t})$$
$$= -\log \frac{\exp\big((\mathbf{x}_{t-1}^L)^\top \mathbf{v}_{x_t}/\tau\big)}{\sum_{x \in V} \exp\big((\mathbf{x}_{t-1}^L)^\top \mathbf{v}_x/\tau\big)}.$$

**Definition D.2** (InfoNCE loss). Let $q \in \mathbb{R}^d$ be a query, let $k^+ \in \mathbb{R}^d$ be a positive key, and let $\mathcal{K}$ be a finite set of candidate keys that contains $k^+$ and negatives. Given a similarity function $\text{sim}(\cdot, \cdot)$ and temperature $\tau > 0$, the InfoNCE loss is

$$\mathcal{L}_{\text{InfoNCE}} = -\log \frac{\exp\big(\text{sim}(q, k^+)/\tau\big)}{\sum_{k \in \mathcal{K}} \exp\big(\text{sim}(q, k)/\tau\big)}.$$

**Proposition D.3** (Token-level NLL is an InfoNCE instance). _Let_ $\text{sim}(a, b) = a^\top b$. _By setting_

$$q = \mathbf{x}_{t-1}^L, \quad k^+ = \mathbf{v}_{x_t}, \quad \mathcal{K} = \{\mathbf{v}_x\}_{x \in V},$$

_we obtain_ $\mathcal{L}_{NLL} \equiv \mathcal{L}_{InfoNCE}$.

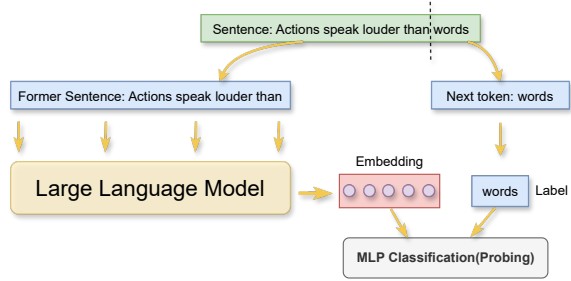

_Figure 5._ Architecture for Probing Next Token Prediction.

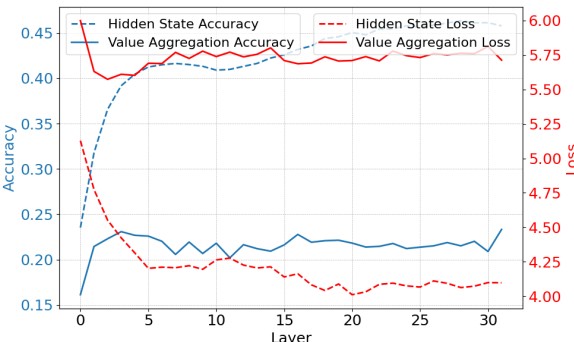

_Figure 6._ Accuracy and Loss in Next Token Prediction Probing.

_Proof._ Substituting the specified $q$, $k^+$, $\mathcal{K}$, and $\text{sim}$ into $\mathcal{L}_{\text{InfoNCE}}$ yields

$$-\log \frac{\exp\big((\mathbf{x}_{t-1}^L)^\top v_{x_n}/\tau\big)}{\sum_{x \in V} \exp\big((\mathbf{x}_{t-1}^L)^\top v_x/\tau\big)},$$

which matches $\mathcal{L}_{\text{NLL}}$. $\square$

This equivalence clarifies the supervision signal for HS: it directly trains $x_{t-1}^L$) to align with the true next token. Therefore, pooling hidden states into a single sentence vector inherits a representation space shaped by _token-level discrimination_, not by sentence-level similarity.

**Next-Token Probing.** To validate whether hidden states encode the immediate next token better than value aggregation, we design a probing task where the model predicts the next token given a sentence prefix, shown in Figure 5. This task directly aligns with the training objective of autoregressive modeling. As shown in Figure 6, we find that for next-token prediction, hidden states outperform VA by 15–20 accuracy points, which reflects the direct optimization of hidden states for next-token prediction.

# E. Weight value vectors better support predicting subsequent tokens

The probe uses $\alpha_{t+k, \cdot}^{l,h}$, which depends on the query at position $t+k$. This is informative for analysis, but not available

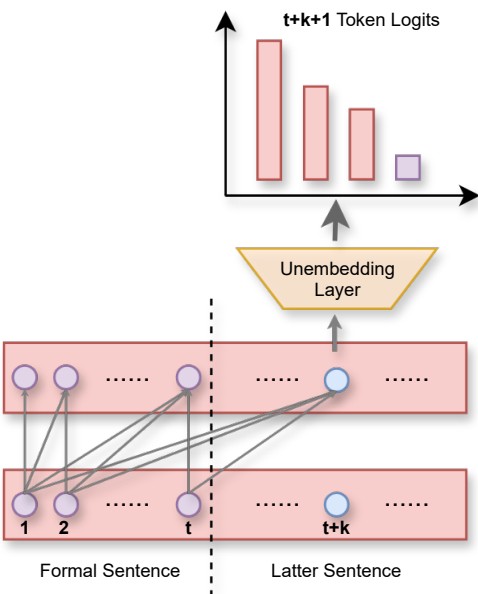

**Figure 7.** Logit Lens probing for predicting subsequent tokens.

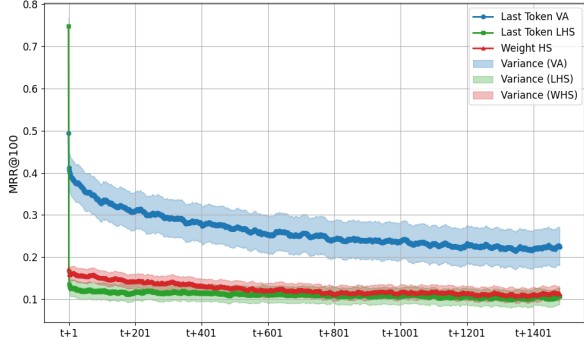

**Figure 8.** MRR@100 for subsequent-token prediction using weighted value aggregation, weighted hidden-state pooling, and last-hidden-state pooling.

at inference time when we want a single embedding from the observed sequence. This motivates weights $\pi$ that can be computed from the prefix alone.

**Empirical Evaluation: Logit Lens Probing.** We use a probing task that measures whether a sentence representation captures information relevant to *long-range continuation*. We adopt a logit-lens style evaluation that maps intermediate representations to vocabulary logits using the model's own unembedding. Unlike standard next-token evaluation, we probe tokens beyond $t+1$ to isolate whether an embedding retains information that remains predictive deeper into the continuation.

**Task Setup.** As shown in Figure 7, given a sequence $x_1, \ldots, x_N$, we select a prefix length $t$. We run the model while restricting attention to the prefix tokens $x_1, \ldots, x_t$, and evaluate predictions for continuation positions $t+k$ with $k \geq 1$. At each continuation position $t+k$, the attention output at layer $l$ and head $h$ is a weighted sum over prefix value vectors:

$$\mathbf{z}_{t+k}^{l,h} = \sum_{j=1}^{t} \alpha_{t+k,j}^{l,h} \, \mathbf{v}_j^{l,h} \in \mathbb{R}^{d_h},$$

$$\mathbf{a}_{t+k}^{l} = \text{Concat}\left(\mathbf{z}_{t+k}^{l,1}, \ldots, \mathbf{z}_{t+k}^{l,H}\right) \mathbf{W}_O^l,$$

where $\alpha_{t+k,j}^{l,h}$ are attention weights computed from the query at position $t+k$ and prefix keys. We then apply the unem-

bedding to obtain a distribution over the vocabulary:

$$p(\hat{x}_{t+k+1} \mid x_1, \ldots, x_t) = \frac{\exp\left((\mathbf{a}_{t+k}^l)^\top \mathbf{v}_{\hat{x}_{t+k+1}} / \tau\right)}{\sum_{x \in V} \exp\left((\mathbf{a}_{t+k}^l)^\top \mathbf{v}_x / \tau\right)},$$

where $\mathbf{v}_x$ is the output token embedding and $\tau$ is a temperature. This probe directly connects to Weighted VA: $\mathbf{z}_{t+k}^{l,h}$ is a weighted aggregation of prefix values, and the weights vary with the continuation position.

**Metrics.** We use mean reciprocal rank at 100 (MRR@100), which measures where the ground-truth token appears in the top-100 predictions. For $M$ instances,

$$\text{MRR@100} = \frac{1}{M} \sum_{i=1}^{M} \frac{1}{\text{rank}_i},$$

where $\text{rank}_i$ is the rank of the ground-truth token if it appears within the top 100 (and contributes 0 otherwise). We compute this score for each layer $l$,

$$\text{MRR@100}_l = \frac{1}{M} \sum_{i=1}^{M} \frac{1}{\text{rank}_i^{(l)}},$$

and report the best layer,

Final MRR@100 $= \max\left(\text{MRR@100}_0, \ldots, \text{MRR@100}_L\right),$

to capture where each representation is most predictive.

**Experimental Setup.** We evaluate on LongBench-v2 (1600 examples), truncating sequences to 2000 tokens to fit memory. We sample prefixes with lengths $t \in [50, 150]$ and probe multiple continuation offsets $k$. We compare three representations: (1) VA with uniform averaging, (2) last-hidden-state pooling (LHS), and (3) weighted hidden-state pooling, where hidden states are reweighted using attention-derived token importance. All methods share the same backbone; only the aggregation strategy changes.

*Table 6.* Finetuned embedding model performance evaluated on various tasks from the MTEB benchmark.

| Model | Dim | Backbone | Clustering | | | Retrieval | | | STS | | | Classification | | | Reranking | | Avg. |
|---|---|---|---|---|---|---|---|---|---|---|---|---|---|---|---|---|---|
| | | | Bior. | Medr. | Twen. | SciF. | NFCo. | Argu. | STS17 | SICK-R | STSB. | Bank. | Emot. | Spri. | Stac. | SciD. | |
| **VA** | 4096 | Llama-2 | 32.45 | 28.65 | 27.95 | 52.41 | 23.52 | 44.26 | 74.08 | 61.49 | 61.72 | 75.19 | 39.54 | 73.75 | 41.51 | 76.70 | 50.94 |
| **Finetuning Methods** | | | | | | | | | | | | | | | | | |
| **Finetune-MP** | 4096 | Llama-2 | 34.41 | 31.19 | 36.69 | 71.47 | 35.56 | 48.81 | 88.19 | 77.71 | 82.83 | 85.77 | 50.55 | 92.28 | 47.87 | 74.60 | 61.28 |
| **Finetune-VA** | 4096 | Llama-2 | 33.30 | 29.97 | 38.26 | 71.86 | 38.21 | 46.88 | 89.76 | 79.97 | 84.62 | 82.92 | 49.32 | 94.67 | 49.29 | 80.98 | 62.14 |
| **Finetune-VA (Attention Only)** | 4096 | Llama-2 | 34.12 | 31.02 | 39.64 | 70.53 | 31.12 | 51.75 | 88.57 | 77.63 | 82.56 | 85.37 | 47.24 | 90.07 | 48.20 | 79.70 | 61.25 |
| **Finetune-MP** | 4096 | Qwen-3 | 33.85 | 30.82 | 40.04 | 73.36 | 29.57 | 52.62 | 89.11 | 77.59 | 85.11 | 85.97 | 48.53 | 87.45 | 50.06 | 75.73 | 61.42 |
| **Finetune-VA** | 1024 | Qwen-3 | 32.67 | 29.84 | 40.20 | 70.95 | 28.97 | 51.32 | 88.98 | 76.95 | 83.44 | 85.24 | 48.78 | 90.86 | 49.03 | 73.81 | 60.79 |
| **Finetune-VA (Attention Only)** | 1024 | Qwen-3 | 31.74 | 29.21 | 38.87 | 70.42 | 29.13 | 53.06 | 88.59 | 76.20 | 82.38 | 84.60 | 46.93 | 90.25 | 48.68 | 74.16 | 60.30 |
| **LLM-based Pretrained Embedding Models** | | | | | | | | | | | | | | | | | |
| **LLM2Vec** | 4096 | Llama-2 | 34.81 | 31.37 | 51.04 | 77.30 | 40.33 | 56.53 | 90.63 | 83.01 | 88.72 | 88.17 | 51.71 | 96.83 | 51.02 | 84.03 | 66.11 |
| **VA (Llm2vec)** | 4096 | Llama-2 | 33.87 | 30.73 | 47.82 | 74.95 | 29.51 | 56.98 | 88.48 | 81.12 | 87.07 | 83.32 | 52.75 | 97.07 | 52.31 | 83.02 | 64.21 |
| **Qwen3-Embedding-0.6B** | 1024 | Qwen-3 | 39.98 | 36.76 | 51.20 | 70.65 | 36.06 | 70.65 | 93.32 | 84.69 | 91.23 | 85.19 | 59.77 | **97.48** | 54.22 | 86.78 | 68.43 |
| **VA (Qwen3-Embedding-0.6B)** | 1024 | Qwen-3 | 39.23 | 36.05 | 49.84 | 68.21 | 32.74 | 67.02 | 89.67 | 80.17 | 87.14 | 84.34 | 59.46 | 96.50 | 54.17 | 85.90 | 66.46 |
| **Qwen3-Embedding-8B** | 4096 | Qwen-3 | **44.57** | **40.56** | **63.40** | **78.61** | **41.39** | **76.30** | **95.82** | **88.46** | **93.59** | **89.71** | 62.82 | 97.46 | **58.86** | **89.96** | **72.97** |
| **VA (Qwen3-Embedding-8B)** | 1024 | Qwen-3 | 44.29 | 39.97 | 60.40 | 75.16 | 34.48 | 72.71 | 94.15 | 83.94 | 91.00 | 89.40 | **63.93** | 96.06 | 58.19 | 88.76 | 70.89 |

*Table 7.* Different layers performance evaluated on various tasks from the MTEB benchmark.

| Selected Layers | Dim | Backbone | Clustering | | | Retrieval | | | STS | | | Classification | | | Reranking | | Avg. |
|---|---|---|---|---|---|---|---|---|---|---|---|---|---|---|---|---|---|
| | | | Bior. | Medr. | Twen. | SciF. | NFCo. | Argu. | STS17 | SICK-R | STSB. | Bank. | Emot. | Spri. | Stac. | SciD. | |
| **0-20%** | 4096 | Llama-2 | 20.54 | 21.00 | 18.00 | 31.50 | 9.69 | 23.49 | 72.62 | 60.31 | 58.48 | 68.24 | 36.66 | 67.34 | 40.26 | 65.46 | 42.40 |
| **0-50%** | 4096 | Llama-2 | 23.69 | 23.39 | 20.25 | 30.55 | 8.37 | 27.2 | 72.85 | 59.73 | 54.16 | 71.46 | 40.33 | 53.61 | 40.34 | 68.17 | 42.44 |
| **0-100%** | 4096 | Llama-2 | 31.69 | 28.30 | 26.34 | 51.28 | 21.00 | 42.75 | 74.51 | 61.47 | 60.94 | 73.89 | 39.58 | 71.42 | 41.63 | 76.16 | 50.07 |
| **20-50%** | 4096 | Llama-2 | 23.64 | 23.31 | 20.54 | 28.89 | 8.02 | 27.40 | 72.84 | 59.46 | 53.62 | 72.64 | **40.78** | 50.37 | 40.16 | 68.10 | 42.13 |
| **50-85%** | 4096 | Llama-2 | 32.23 | **29.16** | 30.25 | 54.29 | 24.09 | 44.84 | 75.46 | 62.19 | 62.66 | 76.17 | 40.73 | 73.87 | 42.27 | 77.16 | 51.81 |
| **50-100%** | 4096 | Llama-2 | **32.45** | 28.65 | 27.95 | 52.41 | 23.52 | 44.26 | 74.08 | 61.49 | 61.72 | 75.19 | 39.54 | 73.75 | 41.51 | 76.70 | 50.94 |
| **85-100%** | 4096 | Llama-2 | 31.21 | 27.98 | 25.92 | 52.02 | 21.41 | 43.21 | 72.41 | 59.81 | 59.14 | 73.63 | 37.47 | 72.02 | 40.03 | 75.56 | 49.42 |

**Results.** Figure 8 highlights a qualitative shift between predicting the immediate next token and predicting subsequent tokens. For $k=1$, LHS performs best (MRR@100 > 0.7), consistent with the fact that the final-layer representation at the last position is directly optimized for next-token prediction. VA is lower (around 0.5), and weighted hidden-state pooling performs worst (MRR@100 < 0.2).

For $k \geq 2$, VA becomes consistently more effective. Across $k=2$ and $k=3$, VA improves over both baselines by roughly 10–15 MRR@100 points and degrades more slowly as the offset increases. In contrast, LHS drops sharply after $k=1$, suggesting that it is less suitable as a summary for longer-range continuation. These observations support the view that aggregating value vectors retains information that remains predictive beyond the next token, which is aligned with our motivation for value-based pooling.

## F. VA Result for Current Embedding Model

The concrete result for finetuning the model can be seen in Table 6.

**Experimental Setup** We study whether sentence-embedding can also improve fine-tuning value aggregation (VA). In many embedding pipelines, models are trained with a contrastive objective defined on a pooled representation of the last hidden state. If such training also improves value-based representations, then standard retrieval-style training

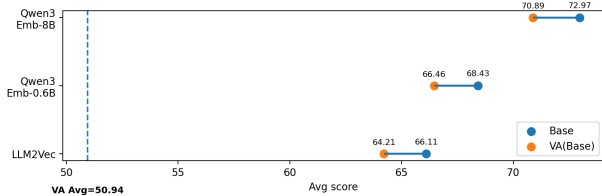

*Figure 9.* Base vs. +VA (VA baseline shown as dashed line).

would implicitly benefit VA, without requiring any change to the loss or optimization procedure.

To test this, we consider two widely used embedding models built on the same backbone families studied above: LLM2VEC (LLaMA-2 backbone) and QWEN3-EMBEDDING (Qwen-3 backbone). We replace each model's original pooling scheme with VA and evaluate the resulting embeddings. For QWEN3-EMBEDDING, we include both the 0.6B and 8B variants. This choice covers both standard multi-head attention (LLaMA-2) and grouped-query attention (Qwen-3), aligning the evaluation with our earlier setting.

**Results and Discussion** Figure 9 shows that contrastive embedding training consistently improves VA across both LLaMA-2 and Qwen-3 backbones. After fine-tuning, VA is within roughly 2 points of last-hidden-state pooling, despite the objective being defined on the final-layer output. This

Table 8. Performance of different signals evaluated on various tasks from the MTEB benchmark.

| Model | Dim | Backbone | Clustering | | | Retrieval | | | STS | | | Classification | | | Reranking | | Avg. |
|---|---|---|---|---|---|---|---|---|---|---|---|---|---|---|---|---|---|
| | | | Bior. | Medr. | Twen. | SciF. | NFCo. | Argu. | STS17 | SICK-R | STSB. | Bank. | Emot. | Spri. | Stac. | SciD. | |
| Query Agg. | 4096 | Llama-2 | 24.42 | 23.32 | 17.79 | 1.78 | 1.84 | 22.3 | 65.19 | 55.65 | 53.05 | 69.74 | 35.17 | 47.51 | 36.91 | 64.71 | 37.10 |
| Key Agg. | 4096 | Llama-2 | 22.18 | 22.71 | 16.99 | 0.93 | 1.85 | 19.08 | 64.75 | 57.11 | 55.83 | 71.05 | 36.24 | 49.62 | 36.96 | 63.80 | 37.08 |
| Activation Agg. | 11008 | Llama-2 | 29.30 | 27.76 | **31.14** | 44.17 | 14.62 | 39.91 | 65.11 | 55.50 | 54.05 | 73.13 | 38.21 | 54.01 | 40.16 | 69.72 | 45.49 |
| Value Agg. (VA) | 4096 | Llama-2 | **32.45** | **28.65** | 27.95 | **52.41** | **23.52** | **44.26** | **74.08** | **61.49** | **61.72** | **75.19** | **39.54** | **73.75** | **41.51** | **76.70** | **50.94** |

Table 9. VA performance with different Qwen3 backbone sizes evaluated on various tasks from the MTEB benchmark.

| Model | Dim | Backbone | Clustering | | | Retrieval | | | STS | | | Classification | | | Reranking | | Avg. |
|---|---|---|---|---|---|---|---|---|---|---|---|---|---|---|---|---|---|
| | | | Bior. | Medr. | Twen. | SciF. | NFCo. | Argu. | STS17 | SICK-R | STSB. | Bank. | Emot. | Spri. | Stac. | SciD. | |
| VA (Full) | 1024 | Qwen3-0.6B | 26.76 | 25.78 | 22.35 | 40.11 | 13.62 | 33.75 | 72.36 | 61.86 | 59.95 | 72.15 | 33.39 | 69.66 | 41.83 | 68.29 | 45.85 |
| VA (Half) | 1024 | Qwen3-0.6B | 26.83 | **25.86** | **22.88** | 40.43 | 14.17 | 33.92 | 72.71 | **62.06** | 60.19 | 72.56 | 33.27 | 70.61 | 42.01 | 68.55 | 46.15 |
| VA (Full) | 1024 | Qwen3-4B | 26.89 | 24.66 | 20.15 | 53.66 | 18.19 | 38.49 | 70.97 | 60.43 | 59.42 | 74.49 | 33.43 | 79.39 | 44.04 | 69.96 | 48.16 |
| VA (Half) | 1024 | Qwen3-4B | 26.56 | 24.60 | 19.54 | 53.42 | 18.21 | 38.52 | 71.68 | 60.75 | 60.03 | 74.87 | 33.21 | 79.63 | 44.28 | 69.89 | 48.23 |
| VA (Full) | 1024 | Qwen3-8B | 26.85 | 24.65 | 20.59 | **58.37** | 18.69 | 41.41 | 71.70 | 60.38 | 60.77 | 75.72 | 33.20 | 81.92 | 44.72 | **70.45** | 49.24 |
| VA (Half) | 1024 | Qwen3-8B | **26.94** | 24.29 | 21.92 | 58.29 | 18.80 | 41.49 | 71.83 | 60.47 | 60.91 | 75.71 | 32.91 | 82.10 | **44.79** | 70.44 | 49.35 |
| VA (Full) | 1024 | Qwen3-14B | 25.84 | 24.31 | 18.75 | 58.05 | **19.52** | 41.71 | **75.29** | 60.99 | **62.01** | 75.87 | **34.67** | 82.19 | 44.76 | 70.25 | 49.57 |
| VA (Half) | 1024 | Qwen3-14B | 25.69 | 24.60 | 19.82 | 57.79 | 19.47 | **41.86** | 75.19 | 60.99 | 61.98 | **76.04** | 34.41 | **82.32** | 44.75 | 70.23 | **49.65** |

Table 10. Encoding time and peak GPU memory usage on the SciDocsRR dataset.

| Method | Encode (minutes) | Peak GPU Memory (MiB) |
|---|---|---|
| MetaEOL | $\sim 279$ | $\sim 13738$ |
| AlignedWVA | $\sim 40$ | $\sim 13762$ |

suggests that the supervision signal induced by contrastive training propagates through the attention computation and improves value representations, making VA compatible with existing embedding training pipelines without modifying the loss.

## G. Detailed Analysis of Layer Selection

The results in Table 7 show that, for LLaMA-2-7B, VA is relatively stable with respect to layer selection once deeper layers are included. In particular, layer groups from the later part of the model, especially those covering 50–100% of the model depth, yield consistently strong results across tasks. This indicates that VA does not depend on a narrowly selected layer subset, but instead can use value representations from a reasonably broad range of deeper layers.

This observation is consistent with prior findings in (Queipo-de-Llano et al., 2025), which report that compression valleys often appear around 20–85% of the layer depth across different LLM families. These layers are likely to provide a useful trade-off: they are deep enough to contain contextualized semantic information, but not restricted to only the final layers, where representations may become more tied to next-token prediction. Another study (Wang et al., 2023) suggests that deeper layers, especially those in the 50–100% range, are more related to extracting and using task-relevant information.

Based on these prior findings and our empirical results, we claim to use layers from 50–85% of the model depth as a practical layer-selection rule. This choice keeps the selected layers within the empirically strong region while avoiding the most final layers only. It also reduces the need for model-specific layer tuning, making VA easier to apply across different backbones.

## H. Performance of Different Signals in LLM Transformer Blocks

The design of VA is based on the intuition that value vectors are the main carriers of the information passed through the attention operation. In self-attention, queries and keys are mainly used to compute matching scores, while values provide the content that is weighted and aggregated. Therefore, if attention can be viewed as selecting and mixing information from the input sequence, value vectors are a natural source for sentence-level representation learning. To further test this design choice, we compare VA with other signals from the Transformer block in Table 8. Specifically, we evaluate aggregation based on query vectors, key vectors, value vectors, and activation outputs.

From Table 8, VA achieves the best performance among all aggregated signals. Query aggregation and key aggregation show similar representation quality, which is consistent with their paired roles in attention-based matching. However, both perform poorly on retrieval tasks. Activation aggregation performs better than query and key aggregation, but it has a much higher dimensionality, which increases storage and computation costs in downstream use. In contrast, VA gives the best average performance while keeping the same 4096-dimensional representation size as query and key aggregation.

## I. VA Scaling Analysis

We further study whether VA benefits from larger back-bones. The results in Table 9 show a clear scaling trend: VA performance improves as the Qwen3 backbone size increases from 0.6B to 14B. This suggests that value representations become more effective as the base model gains stronger language modeling and semantic representation ability. The results on QWEN3-EMBEDDING-0.6B and QWEN3-EMBEDDING-8B in Table 6 provide additional support for this trend. These results indicate that VA can benefit from both larger pretrained models and contrastive embedding fine-tuning.

## J. Runtime Comparison

MetaEOL uses 8 prompts and aggregates embeddings from 8 forward passes, making it computationally more expensive than VA variants, which require only a single forward pass. In contrast, the peak GPU memory usage of the two methods is largely comparable. Specifically, we conduct experiments on the SciDocsRR dataset with 89,133 samples, using a batch size of 1 on a single NVIDIA H20 GPU with 97,871 MiB of memory. All experiments are run under Py-Torch 2.9.1+cu128 with eager attention. Table 10 reports the embedding generation time and peak GPU memory usage under the same hardware and software settings.

