# OpenReview forum: "LLM-based Embeddings: Attention Values Encode Sentence Semantics Better Than Hidden States"
_ICML.cc/2026/Conference — ICML 2026 regular_

### Official Review · Reviewer_hDaW · 2026-02-23

**Soundness:** 3
**Presentation:** 3
**Significance:** 3
**Originality:** 3
**Overall Recommendation:** 4
**Confidence:** 4

**Summary:**

This paper shows that aggregating value vectors from attention layers (rather than hidden states) produces better training-free LLM sentence embeddings. It proposes VA (Value Aggregation) and AlignedWVA (using W_O projection and last-token attention weights). Training-free VA outperforms other prompt-free methods and rivals MetaEOL (which needs 8 forward passes) on MTEB. AlignedWVA achieves state-of-the-art among training-free LLM embeddings. The work is grounded in truth-conditional semantics: a sentence's meaning is captured by its influence on continuations, which is determined by value vectors.

**Compliance With Llm Reviewing Policy:**

Affirmed.

**Final Justification:**

The addition of LLaMA-3-8B results strengthens the generality claim. The computational cost numbers (8 A100s, 42 hours for full MTEB) provide useful context for reproducibility.

The circular validation concern stands: the segment-matching experiment tests exactly what the theory predicts VA excels at (continuation prediction). An evaluation on tasks orthogonal to the theoretical motivation would have been more convincing. The SciFact 210x gap between HS and VA suggests the baselines are too weak to meaningfully benchmark improvements.

14 of 56+ MTEB tasks is a limited evaluation for broad claims about training-free sentence embeddings. The authors acknowledged the compute constraint, which is fair.

The core insight that attention values encode better sentence semantics than hidden states is novel and worth investigating further. The paper is well written and the analysis is thorough within its scope. Rating unchanged at 4.

**Key Questions For Authors:**

1. Why does VA underperform on classification tasks (Banking77, EmotionClassification)? Is there a systematic explanation within the truth-conditional framework?

2. How does VA scale with model size? Would value vectors from 70B models be even more informative?

3. Is there a principled way to determine optimal layers without task-specific validation data?

4. What is the computational overhead of extracting value vectors versus reading hidden states?

5. Why were only 14 of 56+ MTEB tasks selected? What criteria determined the subset?

**Limitations:**

Key limitations include: (1) model-specific and task-specific layer selection undermines the training-free claim; (2) evaluation on only 14 of 56+ MTEB tasks; (3) only two backbone models tested; (4) underperformance on classification tasks unexplained; (5) preliminary fine-tuning results without SOTA comparisons.

**Strengths And Weaknesses:**

## Strengths

1. The core insight is compelling and novel: value vectors capture sentence semantics better than hidden states because hidden states are optimized for next-token prediction, not sentence-level meaning. The illustrative example in Section 3.1 (merger vs patient sentences sharing continuation space) makes this concrete and intuitive.

2. The segment-matching experiment (Figure 2) on LongBench v2 provides empirical evidence for the hypothesis, though the experiment design closely mirrors the theory it validates.

3. VA is remarkably simple - mean-pool value vectors across selected layers and tokens. No prompts, no extra forward passes, no training. Compared to MetaEOL (8 forward passes), this is far more practical.

4. AlignedWVA achieves SOTA among training-free LLM embeddings while being computationally cheaper than MetaEOL. Table 1 shows it outperforms MetaEOL on both LLaMA-2 and Qwen-3 backbones.

5. The truth-conditional semantics framework (Section 3.2) is a creative bridge between philosophy of language and NLP.

## Weaknesses

1. Layer selection for VA is model-specific and task-specific: layers 20-27 for LLaMA-2, layers 26-31 for Qwen-3. This requires held-out validation, undermining the 'training-free' narrative.

2. Table 1 shows VA underperforms on classification tasks: Banking77 at 76.15 vs 81.79 (EE, LLaMA-2). Value vectors do not universally outperform hidden states.

3. Only 14 of 56+ MTEB tasks are evaluated. Selecting a subset risks cherry-picking favorable tasks.

4. The extreme retrieval improvements (SciFact: HS=0.26 vs VA=54.58, a 210x difference) are suspicious, suggesting hidden-state baselines are pathologically broken rather than VA being particularly good.

5. Only two backbone models tested (LLaMA-2-7B, Qwen-3-8B). Two models is too few for general claims about attention value vectors.

6. Fine-tuning results are self-described as 'preliminary' with no comparison to state-of-the-art fine-tuned embedding models (GTE, NV-Embed, Jina Embeddings).

7. The segment-matching experiment risks circular reasoning: the proxy task tests continuation prediction, which is exactly what the theory predicts VA should excel at.

## Missing References

1. Massive Values in Self-Attention (2024, OpenReview) - shows attention layers exhibit concentrated massive values in Q and K but not V.
2. LLM2Vec (BehnamGhader et al., 2024) - comparison with bidirectional attention adaptation would strengthen the paper.
3. NV-Embed (Lee et al., 2024) - state-of-the-art embedding model, relevant for fine-tuning comparison.

---

> ### Author Rebuttal · Authors · 2026-03-31
>
> >W1: Layer selection for VA is model-specific and task-specific … undermining the training-free …
>
> A1: Thank you. We would like to clarify that “training-free” means that VA does not require any parameter update. In addition, we show that VA (Llama-2) is not sensitive to the layer selection, as long as deeper layers are chosen (specifically, from 50–100% of layer depth, i.e. 32 layers in Llama-2-7B). This aligns with empirical studies on the roles of deep layers in LLMs [1],[2].
>
> |Selected Layers | 0-20% |  0-50%|  20-50% |50-85%|85-100%|50-100%| Heldout-Selection|
> | --- | --- | --- | --- | --- | ---  | --- | --- |
> |Avg. | 42.40   | 42.44 | 42.13| 51.81|49.42| 50.94 | 52.25|
>
>  [1] Attention Sinks and Compression Valleys in LLMs are two sides of the same coin  (ICLR‘26)
>
>  [2] Label Words are Anchors: An Information Flow Perspective for Understanding In-Context Learning (EMNLP'23)
>
> >W2:  VA underperforms on classification tasks: Banking77 at 76.15 vs 81.79 (EE, LLaMA-2).
>
> A2: Thank you. We agree that plain VA is not superior to all variants of HS-based methods. For example, EE performs better on certain classification tasks, but at an additional cost. It is noteworthy that, similar to how various methods have been developed from HS, we demonstrate that multiple variants based on VA can be developed. The VA-based variants generally offer more advantages over their HS counterparts.
>
> >W3: Only 14 of 56+ MTEB tasks are evaluated.
>
> A3: Thank you. The reason for the use of 14 datasets is the computational cost of running on the full set, which requires 8 A100 cards and runs for 42 hours. The subset is also recommended by LLM2Vec, as it covers the major MTEB task categories. To provide more perspectives, our evaluation has already been more comprehensive compared to MetaEOL, PromptEOL, which were conducted only on the STS subset.
>
> >W4: …hidden-state baselines are pathologically broken rather than VA being particularly good.
>
> A4: Indeed, HS, by nature, is not suitable for capturing sentence semantics. However, it is noted that VA-based methods are also strong. For example, AlignedWVA (FutureEOL, Llama2) obtains 54.98, even better than LLM2Vec (Uni+SimCSE, LLama2-7B), which obtains the results of 52.03 as shown in Table 5 in LLM2Vec paper.
>
> >W5: Only two backbone models tested (LLaMA-2-7B, Qwen-3-8B).
>
> A5: Thank you. We will add the backbone of LLaMa-3-8B to the final version.
>
> | Model | Clus. | Ret. | STS | Cls | Rerank | Avg |
> | --- | --- | --- | --- |--- | --- | --- |
> | HS | 15.38 | 16.31 | 44.76 | 51.78 | 46.25 | 34.09 |
> | VA| 28.31 | 40.76 | 63.19 | 59.04 | 56.90 | 49.12 |
> |AlignedWVA (FutureEOL) | 29.92 | 42.88 | 75.64 | 64.40 | 59.52 | 54.11 |
>
> >W6: ... no comparison to SOTA fine-tuned embedding models (GTE, NV-Embed).
>
> A6: Thank you. We would like to clarify that the goal of Section 7 is not to compete with fully developed SOTA embedding systems that are trained with much larger proprietary or specially constructed pretraining datasets, which have much more training samples than our training set (E5). Our goal is much narrower: to test whether the value space remains beneficial under the same backbone, training data, and training setup. Just as training embedding models based on HS has been extensively explored, many directions remain to be investigated with VA in future work.
>
> >W7. The proxy task tests continuation prediction, which is exactly what the theory predicts VA should excel at.
>
> A7: Thank you. The proxy task is there to verify the hypothesis (the theory). As such, it is our intention to design the task that is consistent with the hypothesis.
>
> >W8: Missing references
>
> Ans: LLM2Vec and NV-Embed are already cited in our draft. We will include the paper “Massive Values in Self-Attention” in the final version.
>
> >Q1: Why does VA underperform on classification tasks …? Is there a systematic explanation…?
>
> Ans: Please refer to W2. Plain VA is actually better than plain HS (without special prompts) in a fair comparison. To be fair, EE, MetaEoL, PromptEOL with HS should be compared to WVA or AlignedWVA (in Table 2).
>
> > Q2: How does VA scale with model size?
>
> Ans: We are running additional experiments and hope to get the results soon.
>
> > Q4: What is the computational overhead of extracting value vectors versus reading hidden states?
>
> Ans: There is no additional overhead if we compare VA against HS, as both require only a single forward pass. Specifically, we sampled 6172 sentences from SciDocs, and measured that the encoding times of VA and HS (batch size of 1 on a single NVIDIA H20 GPU (97,871 MiB) under PyTorch 2.9.1+cu128 with eager attention). Both methods need approximately 2.2 minutes for sentence encoding.
>
> In fact, with a more careful implementation where we conduct early stops at the last selected layers in VA, the encoding time of VA is even less than extracting HS from the final layer.
>
> >Q3&5: optimal VA layers … and the experimental subset.
>
> Ans: Please check answers to W1, and W3.

---

> > ### Author Rebuttal · Reviewer_hDaW · 2026-04-01
> >
> > I read the rebuttal carefully. The addition of LLaMA-3-8B results and computational cost numbers strengthens the paper.
> >
> > The circular validation concern remains. The segment-matching experiment tests continuation prediction, which is exactly what the theory predicts VA should excel at. The response confirms this is intentional. A stronger validation would use tasks orthogonal to the theoretical motivation, such as probing compositional semantics or entailment relationships that do not reduce to continuation modeling.
> >
> > The SciFact result is concerning. HS scores 0.26 vs VA at 54.58, a 210x gap. The response acknowledges HS is not suitable for sentence semantics, which raises the question of why it is included as a baseline at all. If baselines are pathologically weak, improvements may be inflated.
> >
> > 14 of 56+ MTEB tasks is insufficient for claims about training-free sentence embedding methods. The computational cost argument (8 A100s, 42 hours) is noted, but claims should be scoped accordingly.
> >
> > The hypothesis about classification underperformance (VA captures token-level semantics better for STS/retrieval but not categorical distinctions) is speculative without systematic analysis within the truth-conditional framework.
> >
> > The core insight that attention values encode semantics better than hidden states is valuable. Rating unchanged.

---

> > > ### Author Response · Authors · 2026-04-05
> > >
> > > Thank you very much for reading our rebuttal carefully. We also appreciate your thoughtful comments and questions.
> > >
> > > > Q1: … The segment-matching experiment tests continuation prediction, which is exactly what the theory predicts VA should excel at …
> > >
> > > We think there might be some misunderstanding here. The “theory predicts VA should excel at continuation prediction” is not established, yet the hypothesis to be verified. We would like to clarify our logical reasoning as follows:
> > >
> > > -	Hypothesis “VA is better than HS in capturing sentence semantics”.
> > >
> > > -	Premise: Capturing sentence semantics in truth condition means being able to predict continuation (or, segment matching).
> > >
> > > -	Therefore, to verify that **VA is better than HS in capturing sentence semantics**, we verify that **VA is better than HS in segment matching**. The experiment is logically designed for such verification.
> > >
> > > If we understand correctly, you have concern about the premise, meaning “Capturing sentence semantics requires more than segment matching”, then perhaps we can provide additional support by supplementing the motivation section with the results from Table 1, particularly the task of STS (Semantic Textual Similarity), where VA also outperforms HS by large gap (31.2 Points in STS17).
> > >
> > > > Q2: …why it (HS) is included as a baseline at all..
> > >
> > > (Vanilla) HS is included as the baseline because we need head-to-head comparisons between vanilla VA and vanilla HS, which excludes all the enhancement methods via promptings (PromptEOL, MetaEOL) or repeating the original sentence (EE) or applying some special weighting (WHS).
> > >
> > > HS is there not for inflating the improvement, but for a fair comparison, vanilla HS vs vanilla VA. In fact, we focus on the improvement over the stronger versions of HS like PromptEOL or MetaEOL, rather than over vanilla HS. On average, AlignedWVA(FutureEOL) outperforms MetaEOL by 4.15 point on Llama-2 and 4.12 points on Qwen3, with much time advantages. Specifically, the encoding time on the SciDocsRR dataset with 89,133 samples using a batch size of 1 on a single NVIDIA H20 GPU (97,871 MiB) under PyTorch 2.9.1+cu128 with eager attention:
> > >
> > > |Method | Encode (minutes)
> > > | --- | --- |
> > > | MetaEoL | ~ 279 |
> > > | AlignedWVA | ~ 40 |
> > >
> > >
> > > > Q3: 14 of 56+ MTEB tasks is insufficient … claims should be scoped accordingly.
> > >
> > > We would like to clarify several points:
> > > -	The computation cost: with our infrastructure (around 1 H20), according to the statistics, it may take around 10 days to run one round of full 56+ MTEB tasks. For drawing meaningful conclusion, we need to run our method along with the strongest HS-based baseline (MetaEOL). Noted that the original paper of MetaEOL only reports on 7 STS subsets of MTEB, thus we do not have the reference results for 56+ MTEB. Given that MetaEOL needs 7x the encoding time of AlignedWVA, we need (7+1)*10 days = 80 days. This is more than what we can afford.
> > > -	The purpose of proving the potential of VA: The experiments on diverse tasks of 14 datasets are to show the potential of VA, rather than to claim SOTA on MTEB, which is dominated by large embeddings models from big companies. We believe our results are sufficient to prove our points, and extra experiments are only nice to have.
> > >
> > >
> > > > Q4: The hypothesis about classification underperformance … is speculative…
> > >
> > > It is speculative, yet it is not groundless. For STS and Retrieval tasks, we can directly use the embeddings for similarity. In contrast, Clustering tasks require a clustering algorithm (k-means in MTEB) and classification tasks needs a classifier (logistic regression used in MTEB) on top of embeddings. This adds a layer of complexity, and such investigation requires more in-depth and comprehensive analysis, making the scope this paper too broad.
> > >
> > > > Review 5: How does VA scale with model size?
> > >
> > > Thank you for this question. We add the following results to support that VA is enhanced with larger model:
> > >
> > > | Model | Backbone | Clustering | Retrieval | STS | Classification | Reranking | Avg |
> > > |---|---|---:|---:|---:|---:|---:|---:|
> > > | VA(half) | Qwen3-0.6B | 25.19 | 29.51 | 64.99 | 58.81 | 55.28 | 45.85 |
> > > | VA(half) | Qwen3-4B | 23.57 | 36.72 | 64.15 | 62.57 | 57.09 | 48.23 |
> > > | VA(half) | Qwen3-8B | 24.38 | 39.53 | 64.40 | 63.57 | 57.62 | 49.35 |
> > > | VA(half) | Qwen3-14B | 23.37 | 39.71 | 66.05 | 64.26 | 57.49 | 49.65 |
> > >
> > >
> > > In addition, the results in Table 6 in the appendix Qwen-embedding-0.6B and Qwen-embedding-8B provides additional insight on how model size affects VA embeddings. We put the results here for your reference:
> > >  | Model | Backbone | Clustering | Retrieval | STS | Classification | Reranking | Avg |
> > > |---|---|---:|---:|---:|---:|---:|---:|
> > > | VA| Qwen3-embedding-0.6B | 41.71 | 55.99 | 85.66 | 80.1 | 70.04 | 66.46 |
> > > | VA | Qwen3-embedding-8B | 48.22 | 60.78 | 89.70 | 83.13 | 73.48 | 70.89 |
> > >
> > > Thank you again for reading our paper and rebuttal carefully, we hope our reponse can address your concern. We will revise  our paper for clarity.

---

### Official Review · Reviewer_GYp4 · 2026-03-12

**Soundness:** 2
**Presentation:** 3
**Significance:** 2
**Originality:** 3
**Overall Recommendation:** 4
**Confidence:** 5

**Summary:**

This paper challenges the prevailing convention in Large Language Model (LLM) representation learning, which predominantly relies on final-layer hidden states for sentence embeddings. The authors argue that hidden states are optimized for next-token prediction and thus fall short in capturing global sentence semantics. To address this, they propose Value Aggregation (VA) and Aligned Weighted VA (AlignedWVA), which extract and pool attention value vectors across multiple layers and tokens. Evaluated on a subset of the MTEB benchmark, the proposed training-free methods demonstrate highly competitive performance, and preliminary fine-tuning experiments show high parameter efficiency.

**Compliance With Llm Reviewing Policy:**

Affirmed.

**Final Justification:**

I have decided to uphold the recommendation of “weak accept.” Overall, this is a solid empirical paper, and I believe the methodology proposed by the authors is reliable and effective. However, the paper lacks theoretical depth, and the thoroughness of the ablation experiments could be improved; this is why I am not raising the grade.

**Key Questions For Authors:**

1. In Section 7, Finetune-VA shows minor performance drops in clustering and classification tasks compared to Finetune-MP. Could you expand on why the value space might excel at fine-grained semantic matching (STS/Retrieval) but struggle slightly with category-level discrimination?

2. The current layer selection strategy relies heavily on empirical observation per model. Do you have insights into a more generalized, model-agnostic heuristic for layer selection (e.g., consistently taking the last 20% of layers) to make this method more plug-and-play for new architectures?

**Limitations:**

Yes

**Strengths And Weaknesses:**

Strengths
*   **Novel and Inspiring Perspective:** The paper successfully steps outside the "hidden state pooling" echo chamber. By shifting the focus to the attention value space, it opens up a promising new avenue for LLM representation learning.
*   **Exceptional Computational Efficiency:** The foundational VA method is entirely prompt-free and introduces zero additional encoding latency. Furthermore, the AlignedWVA method achieves state-of-the-art performance with a single forward pass, bypassing the heavy computational burden of ensemble-prompt methods like MetaEOL (which requires 8x the inference cost).
*   **Significant Dimensionality Reduction (GQA Synergy):** A highly practical advantage is demonstrated on Grouped-Query Attention (GQA) architectures like Qwen-3. VA generates embeddings at a fraction of the hidden state dimensionality (e.g., 1024 vs. 4096 dimensions) while maintaining strong retrieval performance, drastically reducing downstream storage and compute costs.
*   **Parameter-Efficient Fine-Tuning:** The authors empirically prove that fine-tuning only the attention modules (less than 25% of the parameters compared to full tuning) yields performance on par with fine-tuning the full hidden-state pipeline.

Weaknesses
*   **Lack of Theoretical Depth:** From a purely empirical standpoint, this is a high-quality paper with solid experimental design. However, for the ICML community, it lacks the expected theoretical rigor. The introduction of "Truth-Conditional Semantics" to explain *why* value vectors are superior feels more like a post-hoc philosophical justification rather than a rigorous mathematical or representation-learning foundation. The theoretical connection is somewhat forced and does not provide concrete bounds or convergence proofs.
*   **Arbitrary Selection of Attention Values:** The logical leap from "hidden states are suboptimal" to "value vectors are the definitive solution" is too abrupt. An LLM's Transformer block contains a rich variety of intermediate activations, including Queries, Keys, and Feed-Forward Network (FFN) activations. Recent literature (e.g., *Enhancing Few-Shot Vision-Language Classification with Large Multimodal Model Features*, ICCV 2025) has demonstrated that other internal features can also heavily encode semantic information. Without a systematic ablation study comparing Value vectors against these other internal states, the decision to use Attention Values as the fundamental aggregation element appears somewhat arbitrary.
*   **Limited Exploration of the Upper Bound:** While the empirical finding is genuinely important, the paper leaves the potential ceiling of this approach largely unexplored. The current method applies direct pooling and basic weighting. It is difficult to see how this fundamental step can be significantly advanced in the future, or how one might design pre-training/fine-tuning loss objectives specifically tailored to optimize the "value space" rather than simply applying standard contrastive learning.
*   **Restricted Evaluation Scope:** The evaluation is conducted on a 14-dataset subset of the MTEB benchmark rather than the complete suite. While the authors state this is to maintain category proportions, omitting the full benchmark in a highly competitive subfield leaves room for concerns regarding cherry-picking, especially when assessing generalization.

---

> ### Author Rebuttal · Authors · 2026-03-31
>
> >W1: Lack of theoretical depth
>
> A1: Thank you for your comments. We agree that additional theoretical analysis would strengthen our paper. However, while the connection between HS and next-token representation can be derived analytically (as shown in Appendix D), the relationship between attention values and continuation embeddings is more complex. Therefore, we opted for empirical verification in Section 3. Nevertheless, we welcome any suggestions you may have for future work.
>
> >W2: An LLM’s transformer block contains a rich variety of intermediate activations, including queries, keys, and FFN activations …
>
> A2: Thank you. The use of value aggregation stems from the intuition that while keys are used for search, value vectors are responsible for storing context information. Nevertheless, we agree that a more systematic study would strengthen our work. Therefore, we provide a comparison of VA against other signals within the attention block below:
>
> | Model | Clus. | Ret. | STS | Cls | Rerank | Avg |
> | --- | --- | --- | --- |--- | --- | --- |
> | VA | 31.09 | 42.07 | 66.77 | 63.99 | 59.83 | 52.25 |
> | Activation Agg. | 29.40 | 32.90 | 58.22 | 55.11 | 54.94 | 45.49 |
> | Query Agg. | 21.84 | 8.64 | 57.96 | 50.80 | 50.81 | 37.10 |
> | Key Agg. | 20.60 | 7.28 | 59.28 | 52.30 | 50.38 | 37.08 |
>
> >W3: Limited exploration of upper-bound … how this fundamental step can be significantly advanced in the future....
>
> A3: Although not the main focus of our current work, finetuning-VA demonstrates the potential impact of our new perspective for the study of LLM-based embeddings. Just as training embedding models based on HS has been extensively explored, many directions remain to be investigated with VA—for example, how to train the weighting of AlignedWVA to better capture continuation; how different training strategies such as those in LLM2Vec [1] or Llama2Vec [2] can be applied to VA-based aggregation; and how such results can be extended to multi-modal VLMs for multi-modal or cross-modal retrieval.
>
> [1] LLM2Vec: Large Language Models Are Secretly Powerful Text Encoders (COLM‘24)
>
> [2] LLM2Comp: Learning to Compress: Unlocking the Potential of Large Language Models for Text Representation (AAAI’26)
>
> Recently, we have attempted to finetune VA and the combination weights, which is inspired by AlignedWVA, and the performance is positive, as shown in the following (Llama-2-7b backbone).
>
> |Model | Clus.| Ret. | STS| Cls | Rerank | Avg |
> | --- | --- | --- | --- |--- | --- | --- |
> | Finetune-VA (+weightings) | 36.67 | 56.36 | 83.78 | 77.88 | 66.56 | 64.09 |
> | Finetune-VA | 33.84 | 52.32 | 84.78 | 75.64 | 65.14 | 62.14 |
>
> >W4: Restricted evaluation scope … on a 14-dataset subset of MTEB dataset
>
> A4: Thank you. The main reason for the use of 14 dataset subset is due to the computational cost of running on the full set. As it is noted by LLM2Vec, one full run requires 8 A100 cards and runs for 42 hours. The subset is also recommended by LLM2Vec, as it covers the major MTEB task categories. To provide more perspectives, our evaluation has already been more comprehensive compared to MetaEOL, PromptEOL, which were conducted only on the STS subset.
>
> >Q1: Finetune-VA shows minor performance drops in clustering and classification … Could you expand on why the value space might excel at fine-grained semantic matching (STS/Retrieval)… ?
>
> Ans1: This is an interesting question. Our hypothesis is that VA enables flexible token-level information capture, which facilitates fine-grained semantic matching of STS and Retrieval. Clustering and classification tasks, on the other hand, require finding an optimal combination of features that aligns with predefined categories. As such, further attention should be directed toward studying the mapping function, which has not been optimized in the current finetuning.
>
> >Q2: Insights into a more generalized, model-agnostic heuristic for layer selection.
>
> Ans2: Thank you. Our results below show that for Llama-2-7B, VA is not sensitive to layer selection as long as deeper layers are chosen (specifically, from 50–100% of layer depth, i.e. 32 layers in Llama-2-7B). This aligns with the extensive experiments in Queipo-de-Llano and LeCun's ICLR 2026 paper "Attention Sinks and Compression Valleys in LLMs are two sides of the same coin," which demonstrates that compression valleys appear around layers 20-85% of layer depth across a broad range of LLMs. Another paper from EMNLP'23 Best Papers "Label Words are Anchors: An Information Flow Perspective for Understanding In-Context Learning" states the deeper layer (50-100%) extract and utilize the information. Combining these two views, we find the reasonable performance (50-85%). This is, in general, the rule of thumb that we can apply across a wide range of LLM families.
>
> |Selected Layers | 0-20% |  0-50%|  20-50% |50-85%|85-100%|50-100%| Heldout-Selection|
> | --- | --- | --- | --- | --- | ---  | --- | --- |
> |Avg. | 42.40   | 42.44 | 42.13| 51.81|49.42| 50.94 | 52.25|

---

> > ### Author Rebuttal · Reviewer_GYp4 · 2026-04-04
> >
> > Thank you for the clarification; I will keep my rating.

---

> > > ### Author Response · Authors · 2026-04-07
> > >
> > > Dear Reviewer GYp4,
> > >
> > > Thank you very much for taking the time to read our rebuttal and for confirming that your concerns have been fully resolved. We sincerely appreciate your careful consideration.
> > >
> > > We fully respect your judgment on the final rating. We just wanted to gently note that, since the concerns have now been adequately addressed, we would be deeply grateful if you might consider whether the score could also be updated to reflect your current assessment, if you feel that would be appropriate.
> > >
> > > If helpful, we would also be very happy to provide any further clarification on the paper or our rebuttal. As authors, even a small score adjustment can make a meaningful difference in the final discussion process, so we would truly appreciate any reconsideration.
> > >
> > > Thank you again for your time, thoughtful review, and generous help in improving our paper.

---

### Official Review · Reviewer_8wB8 · 2026-03-13

**Soundness:** 3
**Presentation:** 2
**Significance:** 2
**Originality:** 2
**Overall Recommendation:** 4
**Confidence:** 3

**Summary:**

This paper argues that, for decoder-only LLMs, attention value vectors are better sentence-level features than hidden states for embedding tasks. It proposes Value Aggregation, which mean-pools token value vectors across selected layers, and then extends it to prompt-based weighted variants, especially AlignedWVA, where last-token attention weights and the output projection matrix are used to form aligned sentence representations. Across a 14-task MTEB subset and two backbones, the paper reports that VA is a strong training-free baseline, and that AlignedWVA outperforms the prompt-based MetaEOL baseline while using fewer forward passes. The paper also includes a smaller fine-tuning study suggesting that VA-style pooling can benefit from contrastive training and that an attention-only variant is parameter efficient.

**Compliance With Llm Reviewing Policy:**

Affirmed.

**Final Justification:**

The authors' response clarifies my concerns on this paper. Therefore, I decide to raise my rating to 4

**Key Questions For Authors:**

Please refer to the weaknesses above.

**Limitations:**

No, the limitations are not sufficiently discussed - please refer to the weaknesses above.

**Strengths And Weaknesses:**

**Strengths:**
- The paper targets at a good research question. Most LLM embedding work defaults to hidden states, so examining value vectors as an alternative representation source is a worthwhile direction. The method is simple, easy to implement, and training-free.
- The empirical signal is nontrivial. In Table 1, VA is consistently much stronger than hidden-state pooling baselines, especially on retrieval and reranking, and for LLaMA-2 the selected-layer VA achieves the best average score in the table among the listed methods.
 - Figure 2 is a useful motivational result. The layer-wise segment-matching curves show a clear separation between VA and HS in deeper layers, which makes the paper’s central intuition easier to buy, at least as an empirical observation.
- The AlignedWVA idea is intuitive. Using the last-token attention weights to map back into residual-stream space is a sensible bridge between value-space aggregation and standard embedding spaces.
- The paper compares two fairly different backbones, LLaMA-2 and Qwen-3, including a GQA model, which gives at least some evidence that the phenomenon is not totally model-specific. The method VA remains competitive after contrastive training, and the attention-only variant offers a plausible parameter-efficiency angle.

**Weaknesses:**
- The method proposed in the paper does not show very significant results. Although the idea is quite insightful, it is difficult to apply in practice.
- How does the same method perform when applied to hidden states? Please conduct a detailed experiment.
- The authors state in the paper that "a key limitation of many existing approaches lies in their reliance on the last-layer hidden state as the sentence embedding." This claim does not hold, as it is well known that the last layer is not suitable for embedding.
- The paper keeps emphasizing that AlignedWVA is cheaper than MetaEOL, but there are no actual timing or memory numbers anywhere in the main paper.

---

> ### Author Rebuttal · Authors · 2026-03-31
>
> >W1: The method proposed in the paper does not show very significant results. Although the idea is quite insightful, it is difficult to apply in practice.
>
> A1: Thank you for your comments. We would clarify two points:
>
> First, **training-free VA achieves strong results and is highly practical**. As shown in Table 1, VA consistently outperforms all prompt-free methods by a large margin—more than 10 points on average compared to the best such method (WMP). This is meaningful as it represents a direct comparison under a fair setting. Among prompt-based methods, excluding MetaEOL, the best baseline is EE with an average of only 46.54, substantially lower than VA's 52.25. The comparison with MetaEOL is inherently unfair to VA, as MetaEOL requires 8 forward passes due to the use of 8 different prompts, whereas VA requires only a single pass. Despite this, AlignedWVA still surpasses MetaEOL by 4 points on average with the Qwen backbone. Furthermore, the method is highly practical, as it only requires reading value vectors during the standard forward pass with minimal overhead. This aspect has also been noted by reviewer sifX.
>
> Second, although not the main focus of our current work, **finetuning-VA demonstrates the potential impact of our new perspective for the study of LLM-based embeddings**. Just as training embedding models based on HS has been extensively explored, many directions remain to be investigated with VA—such as learning better weighting schemes for AlignedWVA, designing training objectives based on continuation modeling while preserving the main goals of contrastive learning such as alignment and uniformity, aggregating multi-layer (multi-view) value information more effectively, applying training strategies from methods such as LLM2Vec or LLM2Comp to VA-based aggregation, and extending the same idea to multimodal VLMs for multimodal or cross-modal retrieval.
>
> [1] LLM2Vec: Large Language Models Are Secretly Powerful Text Encoders (COLM‘24)
>
> [2] LLM2Comp: Learning to Compress: Unlocking the Potential of Large Language Models for Text Representation (AAAI’26)
>
> >W2: How does the same method perform when applied to hidden states? Please conduct a detailed experiment
>
> A2: Thank you. We are not entirely certain we follow your meaning. Table 1 shows a direct comparison between VA and HS under equivalent settings. Specifically,  HS represents direct comparison with VA, as both aggregate over the same set of layers. Similarly, HS(full) vs VA (full) and HS (half) vs VA (half) can be considered fair comparison. The remaining baselines—PromptEOL, MetaEOL, and CP—are prompt-based methods with HS. These methods can be compared to the results of PromptEOL and FutureEOL with WVA and AlignedWVA in Table 2. Furthermore, in response to Reviewer sifX's request, we have added plain VA + PromptEOL and FutureEOL, which can be compared to other prompt-based methods.
>
> | Model | Avg |
> | --- | --- |
> | VA | 52.25 |
> | VA (PromptEOL) | 51.67 |
> | VA (FutureEOL) |  50.24|
> | WVA (PromptEOL) | 51.98 |
> | WVA (FutureEOL) | 54.69 |
> | AlignedWVA (PromptEOL) |  52.51|
> | AlignedWVA (FutureEOL) | 54.98 |
>
> >W3: The authors state in the paper that "a key limitation of many existing approaches lies in their reliance on the last-layer hidden state as the sentence embedding." This claim does not hold, as it is well known that the last layer is not suitable for embedding.
>
> A3: Thank you. Although most of the methods (PromptEOL, MetaEOL, LLM2Vec, Qwen3-Embeddings) rely on the last layer hidden states, we agree that the last layer is not suitable for embedding. We will revise the statement to: "A key limitation of many existing approaches lies in their reliance on hidden states for sentence embeddings, which are not optimal for capturing the global sentence semantics." It is worth noting that we have also compared different aggregation strategies for HS—namely HS (full), HS (half), and HS—rather than only using the final layer.
>
> >W4: The paper keeps emphasizing that AlignedWVA is cheaper than MetaEOL, but there are no actual timing or memory numbers anywhere in the main paper.
>
> A4: Thank you. MetaEOL uses 8 prompts and aggregates embeddings from 8 forward passes, making it inherently more time-consuming than VA variants, which require only a single pass. The peak memory usage, however, is mostly the same for both methods. More concretely, we conducted experiments on the SciDocsRR dataset with 89,133 samples using a batch size of 1 on a single NVIDIA H20 GPU (97,871 MiB) under PyTorch 2.9.1+cu128 with eager attention. The end-to-end embedding generation time (encoding time) and peak memory usage under the same hardware and software are as follows:
>
> |Method | Encode (minutes) | Peak GPU Memory (MiB)  |
> | --- | --- | --- |
> | MetaEOL | ~ 279 | ~ 13738 |
> | AlignedWVA | ~ 40 | ~ 13762|

---

### Official Review · Reviewer_sifX · 2026-03-13

**Soundness:** 3
**Presentation:** 3
**Significance:** 4
**Originality:** 3
**Overall Recommendation:** 5
**Confidence:** 4

**Summary:**

This paper argues that, for decoder-only LLMs, commonly used hidden-state-based sentence embeddings are not ideal because the final hidden representations are strongly shaped by the next-token prediction objective. To address this, the authors propose using attention value vectors as the basis for sentence representations, introducing Value Aggregation (VA) as an alternative embedding extraction method. The paper provides both conceptual motivation and empirical evidence that value vectors better capture information relevant to sentence-level semantics and continuation behavior than aggregated hidden states.
The authors evaluate VA on embedding-style benchmarks from a subset of MTEB across models such as LLaMA 2 and Qwen 3, showing that VA consistently and often substantially outperforms hidden-state-based aggregation. They further compare against prior prompting-based embedding methods and report stronger performance. In addition, the paper proposes Aligned Weighted Value Aggregation, which uses the last token’s attention scores to weight value vectors, and adapts Explicit One-word Limitation (EOL) prompting strategies, including variants such as PromptEOL and FutureEOL, to further improve representation quality.
Finally, the paper includes a fine-tuning-based study to further validate the usefulness of value-vector-based representations. Overall, the main contribution of the paper is to highlight attention values as a strong and underexplored source of sentence embeddings in autoregressive LLMs.

**Compliance With Llm Reviewing Policy:**

Affirmed.

**Final Justification:**

The rebuttal addresses my main concerns.

**Key Questions For Authors:**

Q1. In the fine-tuning experiments (Section 7), the hidden-state baseline (Finetune-MP) uses mean pooling over the last layer, while Finetune-VA aggregates value vectors across multiple layers. This asymmetry makes it difficult to attribute the gains purely to using values over hidden states. Part of the benefit could simply come from multi-layer aggregation. Have the authors considered a Finetune-MP variant that also pools hidden states across multiple layers for a fairer comparison?

Q2. The VA baseline scores reported in Table 2 for LLaMA-2 appear to differ from the VA scores for the same model in Table 1. Are these evaluated on different dataset subsets, or does Table 2 report VA (Half) rather than the selected-layer VA?

Q3. For VA, the layer set ∣S∣ appears fixed. Did the authors explore alternative layer subsets or layer-selection strategies, and how sensitive are the results to this choice?

Q4. Could the authors clarify the extent to which the gains from AlignedWVA(PromptEOL/FutureEOL) are due to the prompting strategy itself versus the use of value aggregation? For example, did the authors test PromptEOL/FutureEOL with plain VA in a more isolated way?

**Limitations:**

Yes

**Strengths And Weaknesses:**

Strengths:

S1. The paper is well motivated and builds a clear conceptual case for using attention value vectors instead of hidden states for sentence embeddings in decoder-only LLMs. The central intuition is easy to appreciate, and the paper goes beyond intuition by providing both analytical motivation and empirical validation.

S2. The experimental section is fairly strong overall. The authors evaluate across multiple embedding-style tasks and show that Value Aggregation (VA) consistently improves over hidden-state-based aggregation. The comparisons against prior prompting-based methods also strengthen the case that the proposed approach is competitive beyond baseline setting.

S3. The paper is significant because it highlights a simple but practically useful alternative to standard hidden-state-based embedding extraction. Since sentence embeddings from autoregressive LLMs are widely used, identifying attention values as a stronger representation source could influence future research.

S4. The work also has a reasonable degree of originality. While the method itself is relatively simple, the perspective of motivating and leveraging attention value vectors in this way is novel and insightful. The combination of theoretical intuition, value-based aggregation, aligned weighting, and EOL-style prompting makes the contribution more than a minor implementation tweak.

S5. In terms of presentation, the paper is overall well written, structured, and easy to follow. The narrative is clear, and the progression from motivation to method to evaluation is mostly smooth.

Weaknesses:

W1. A notable soundness concern is that AlignedWVA does not uniformly improve over VA. In particular, compared to VA, AlignedWVA appears substantially weaker on some clustering and retrieval tasks, especially for the LLaMA setting, and this inconsistency is not sufficiently discussed. Since AlignedWVA is presented as a meaningful extension, the paper would benefit from more analysis of when and why it helps or hurts.

W2. Relatedly, the trends across model families are somewhat inconsistent: LLaMA2 seems to benefit more from VA, whereas Qwen appears stronger with AlignedWVA (maybe more expressive space vs VA). This makes it harder to tell whether the proposed improvements are method-driven or partially tied to model-specific properties such as dimensionality or architecture. The paper would be stronger with evaluation on a broader set of model families, including newer models or at least one or two additional families such as Mistral or LLaMA3-class models.

W3. For VA, the set of selected layers ∣S∣ appears fixed, and the paper does not thoroughly examine whether those chosen layers are actually optimal. More ablations over layer subsets and layer-selection strategies would strengthen the evidence that the reported setup is not overly tuned or arbitrary.

W4. It would also be useful to isolate the effect of prompting variants more clearly. In particular, showing PromptEOL/FutureEOL combined with plain VA more explicitly would help disentangle how much of the gain comes from value aggregation itself versus the prompting strategy layered on top of it.

W5. For the fine-tuning-based setup, the comparison may not be entirely apples-to-apples. The hidden-state baseline appears to rely only on the final-layer hidden state, while the proposed value-based approach aggregates over all layers and tokens.

W6. On presentation, although the paper is generally clear, some result tables could be easier to interpret. Currently, the highlighting is mostly done within categories; it would be more reader-friendly to also mark the overall best score for each dataset or setting more explicitly, for example with bold for overall best and a secondary marker for category-wise best.

W7. The presentation of results would also be easier to compare if Qwen and LLaMA results were separated more cleanly. While page limits may explain the current formatting, the combined presentation sometimes makes one-to-one comparison less immediate.

W8. Table 2 reports VA scores only for LLaMA, which also feels somewhat inconsistent with the Values in Table 1.

W9. Section 3.2 on the truth-conditional motivation and empirical verification may be hard to fully grasp from the main paper alone, as key details are deferred to Appendix B. Moving more of this material into the main text, perhaps with a simple illustrative example of how continuation distributions relate to truth conditions, would make the theoretical contribution more accessible.

---

> ### Author Rebuttal · Authors · 2026-03-31
>
> > W 1&2: ...  AlignedWVA does not uniformly improve over VA...evaluation on a broader set of model families...
>
> A 1&2: AlignedWVA is particularly beneficial for GQA-based LLMs such as Qwen-3 and Llama-3. For these backbones, plain VA produces embeddings in 1024 dimensions, which is less expressive than the residual stream space of 4096 dimensions used by AlignedWVA. This expressive advantage outweighs any potential disadvantages (e.g., on certain clustering and retrieval tasks) for GQA backbones. The results for Llama-3-8B below further support this claim:
>
> | Model | Clust. | Ret. | STS | Clf. | Rerank | Avg |
> | --- | --- | --- | --- | --- | --- | --- |
> | HS | 15.38 | 16.31 | 44.76 | 51.78 | 46.25 | 34.09 |
> | VA |28.31|40.76|63.19|59.04|56.90|49.12 |
> | AlignWVA (FutureEOL) | 29.92 | 42.88 | 75.64 | 64.40 | 59.52 | 54.11 |
>
> > W3: ... layer subsets and layer-selection strategies
>
> A3: Thank you. Our results below show that for Llama-2-7B, VA is not sensitive to layer selection as long as deeper layers are chosen (specifically, from 50–100% of layer depth, i.e. 32 layers in Llama-2-7B). This aligns with the extensive experiments in Queipo-de-Llano and LeCun's ICLR 2026 paper "Attention Sinks and Compression Valleys in LLMs are two sides of the same coin," which demonstrates that compression valleys appear around layers 20-85% of layer depth across a broad range of LLMs. Another paper from EMNLP'23 Best Papers "Label Words are Anchors: An Information Flow Perspective for Understanding In-Context Learning" states the deeper layer (50-100%) extract and utilize the information. Combining these two views, we find the reasonable performance (50-85%). This is, in general, the rule of thumb that we can apply across a wide range of LLM families.
>
> |Selected Layers | 0-20% |  0-50%|  20-50% |50-85%|85-100%|50-100%| Heldout-Selection|
> | --- | --- | --- | --- | --- | ---  | --- | --- |
> |Avg. | 42.40   | 42.44 | 42.13| 51.81|49.42| 50.94 | 52.25|
>
> > W4: Isolating the effect of prompting variants
>
> A4: Thank you for this comment. We added plain VA with PromptEOL/FutureEOL, together with the corresponding WVA and AlignedWVA results, to better isolate the contribution of each component.
>
> | Model | Avg |
> | --- | --- |
> | VA | 52.25 |
> | VA (PromptEOL) | 51.67 |
> | VA (FutureEOL) |  50.24|
> | WVA (PromptEOL) | 51.98 |
> | WVA (FutureEOL) | 54.69 |
> | AlignedWVA (PromptEOL) |  52.51|
> | AlignedWVA (FutureEOL) | 54.98 |
>
> In general, as VA exploits mean pooling over all tokens of the original sentence, the prompts have limited impact. On the other hand, we need to use prompts with WVA and AlignedWVA to fuse information to the last token.
>
> > W5: The fine-tuning comparison being not fully apples-to-apples
>
> A5. Thank you. We believe that aggregating hidden states across layers is largely redundant, because the hidden state from layer l-1 is already carried into the next layer through the residual connection. This likely explains why finetuning only the final layer of HS is common practice. However, we agree that adding such an experiment will strengthen our results. Accordingly, we introduce Finetuning-HS (agg), where we aggregate hidden states in the same manner as VA and perform finetuning on Llama-2-7B. The results below confirm our intuition.
>
> | Model | Avg |
> | --- | --- |
> | Finetune-HS (Agg) | 58.89|
> |Finetune-HS (Last layer) | 61.28|
> |Finetune-VA | 62.14|
>
> > W6 & 7: Table readability and separating Qwen/LLaMA results more clearly
>
> A6&7: Thank you very much. We will ensure these issues are addressed in the final version and thoroughly check for better clarity.
>
> > W8: The inconsistency of VA scores in Table 2
>
> A8: Thank you for your comment. To avoid cross-checking across tables, we copied key results from Table 1 to Table 2, with a primary focus on comparing against the best method (MetaEol of Qwen). As for VA, we inadvertently copied the row corresponding to VA (half) instead of VA. We apologize for this mistake and will correct it in the final version. We will also add the VA scores for Qwen to Table 2.
>
> > W9:  the truth-conditional motivation being hard to follow from the main paper alone
>
> A9: Thank you for your comments. We will move the details and illustrative examples from Appendix B to the main paper.

---

> > ### Author Rebuttal · Reviewer_sifX · 2026-04-03
> >
> > The rebuttal addresses my concerns adequately. I am happy to retain my score.

---

> > > ### Author Response · Authors · 2026-04-07
> > >
> > > Dear Reviewer sifX,
> > >
> > > Thank you very much for taking the time to read our rebuttal and for confirming that your concerns have been fully resolved. We sincerely appreciate your careful consideration.
> > >
> > > We are very grateful for your thoughtful feedback, which has helped us improve the clarity of the paper.
> > >
> > > Thank you again for your time and helpful review.

---

### Decision · Program_Chairs · 2026-04-30

**Decision:**

Accept (regular)

**Comment:**

The paper identifies limitations in traditional hidden-state sentence embeddings for decoder-only LLMs and proposes Value Aggregation (VA), an approach that utilizes attention value vectors to better capture sentence-level semantics. During the review process, concerns were raised regarding the restricted evaluation scope and the inconsistent performance of the AlignedWVA variant. In their rebuttal, the authors adequately addressed these issues by providing the requested analysis and clarifying the method's efficiency. Given the empirical results and the satisfactory resolution of the reviewers' concerns, the paper is recommended for acceptance.